EMBO
Molecular Medicine

# Glycine decarboxylase advances IgA nephropathy by boosting mesangial cell proliferation through the pyrimidine pathway

Yi Xiong[1,3], Fang Zeng[1,2,3], Kaiping Luo[2], Li Wang[1], Manna Li[1], Yanxia Chen[1], Tianlun Huang[1], Chengyun Xu[1], Gaosi Xu ![ORCID][1✉] & Honghong Zou ![ORCID][1✉]

## Abstract

**The proliferation of glomerular mesangial cells is a fundamental pathological change in immunoglobulin A nephropathy (IgAN). This study aims to elucidate the mechanisms that affect the proliferation of glomerular mesangial cells. Bioinformatics analysis combined with clinical detection identified the key molecule glycine decarboxylase (GLDC). In vitro experiments revealed that GLDC knockdown reduces the proliferative effect of pIgA on mesangial cells. Pyrimidine metabolism is involved in the proliferation regulation of mesangial cells by GLDC. Additionally, GLDC's regulation of glycolysis in mesangial cells was discovered, which further affects the progression of renal fibrosis and the proliferation of glomerular mesangial cells. Upon knockdown of the key rate-limiting enzymes of pyrimidine metabolism, CAD and DHODH, the overexpression of GLDC lost its regulatory effect on glycolysis. The regulatory mechanisms described above were confirmed by inhibiting GLDC expression in the kidneys in vivo. In conclusion, GLDC upregulates pyrimidine metabolic flux, which subsequently fuels glycolysis to promote mesangial cell proliferation, promoting IgAN progression.**

**Keywords** GLDC; Pyrimidine Metabolism; Glycolysis; Glomerular Mesangial Cells; IgA Nephropathy
**Subject Categories** Immunology; Urogenital System

## Introduction

Immunoglobulin A nephropathy (IgAN) is the most common primary glomerular disease worldwide, with hematuria, proteinuria and hypertension as its main clinical manifestations. Statistics show that 50% of IgAN patients can progress to end-stage renal disease within 20–25 years (Pattrapornpisut et al, 2021). Because the pathogenesis of IgAN is not fully understood, there is still a lack of reliable specific treatment. The deposition of galactose deficient IgA1 (Gd-IgA1) immune complex is the initiating factor in the pathogenesis of IgAN, which activates glomerular mesangial cells to mediate intracellular signal transduction, promotes the of glomerular mesangial cell proliferation, and then leads to kidney injury (Zhao et al, 2022). In addition, glomerular crescent formation is a relatively common histopathological change in IgAN, and kidney biopsy of 20% to 60% of IgAN patients can find glomerular crescent of different degrees (Chen et al, 2022; Lin et al, 2021). Studies have shown that glomerular mesangial cells can recruit macrophage infiltration by secreting inflammatory molecule monocyte chemoattractant protein-1 (MCP-1), thereby affecting the formation of glomerular crescent formation (Lai et al, 2014; Tucker et al, 2024; Urushihara et al, 2020). Therefore, the proliferation of glomerular mesangial cell plays a vital role in the pathological development of IgAN.

Recent studies have also found that metabolic reprogramming is involved in the occurrence and progression of acute kidney injury, diabetic nephropathy, lupus nephritis (LN) and other kidney diseases, which is one of the current research hotspots (Li et al, 2021). In this study, bioinformatics analysis revealed that glycine decarboxylase (GLDC) was positively correlated with fibrocellular crescent formation in IgAN. GLDC belongs to the glycine cleavage system (GCS), which is the main degradation pathway of glycine. With the continuous development of relevant research, GLDC has been found to be associated with the proliferation of various tumor cells, including breast cancer (Xie et al, 2022), neuroblastoma (Alptekin et al, 2019), gastric cancer (Min et al, 2016), lung cancer (Yuan et al, 2019), *etc*. In view of the above studies, it is of great value to further clarify the effect of GLDC in the occurrence and development of IgAN and explore its mechanism of regulating the proliferation of mesangial cells, which is of great value to search for therapeutic targets of IgAN in the future.

Notably, pyrimidine metabolism is one of the main differences in the serum of IgAN patients (Li et al, 2022). Leflunomide inhibits de novo synthesis of pyrimidine and is commonly used in the treatment of rheumatoid arthritis and LN (Zheng et al, 2023). It is worth noting that leflunomide combined with low-dose glucocorticoids in IgAN can achieve similar effects in reducing proteinuria and protecting renal function as full-dose glucocorticoid therapy, and this regimen has fewer adverse effects (Min et al, 2017). Early

[1]Department of Nephrology, The Second Affiliated Hospital, Jiangxi Medical College, Nanchang University, Nanchang, China. [2]Department of Nephrology, Ganzhou People's Hospital, Ganzhou, China. [3]These authors contributed equally: Yi Xiong, Fang Zeng. ✉E-mail: ndefy08027@ncu.edu.cn; ndefy21195@ncu.edu.cn

studies showed that the increase of uracil nucleotides in diabetic glomeruli is related to the thickening of glomerular basement membrane (Cortes et al, 1980), suggesting that pyrimidine metabolism may be related to mesangial cell proliferation. In addition, pyrimidine metabolism is related to cell proliferation, and the key rate-limiting enzymes carbamoyl-phosphate synthetase 2, aspartate transcarbamylase and dihydroorotase (CAD) and dihydroorotate dehydrogenase (DHODH) in pyrimidine synthesis can promote the proliferation of various cancer cells (Yang et al, 2023). However, whether pyrimidine metabolism regulates mesangial cell proliferation in IgAN remains unknown. Qi et al found that activation of glycolysis affected LN development by promoting mesangial cell proliferation in the glomeruli (Qi et al, 2023). In addition, early studies by Zhang et al, found that GLDC promoted the proliferation of non-small cell lung cancer cells by mediating glycolysis and pyrimidine metabolism (Zhang et al, 2012). Notably, several studies have found that pyrimidine metabolism can enhance the occurrence of glycolysis (He et al, 2022; Zhou et al, 2023). The above studies suggest that the regulatory effect of GLDC on the proliferation of glomerular mesangial cells may involve pyrimidine metabolism and glycolysis.

In summary, this study aimed to explore whether GLDC affects the proliferation of glomerular mesangial cells through pyrimidine metabolism and glycolysis pathway, thereby affecting the development of IgAN.

# Results

## GLDC was highly expressed in IgAN glomeruli

The expression profiles of differential genes extracted from GSE141295 were analyzed by WGCNA combined with clinical characteristics (Park et al, 2020) Data ref: (Park et al, 2020). Different modules are represented by clusters and different colors, where green module, brown module, and red module showed lower height values (Fig. EV1A). The correlation analysis between modules and clinical features showed that green modules were significantly positively correlated with Cellular fibrocellular crescent formation ($r = 0.7$, $P = 0.008$) (Fig. EV1B). A total of 384 genes in the green module were further extracted and KEGG (Kyoto Encyclopedia of Genes and Genomes) pathway analysis was performed. The result showed that GLDC was closely related to pathways, including carbon metabolism, glyoxylate and dicarboxylate metabolism, glycine, serine and threonine metabolism, lipoic acid metabolism ($logFC = 1.62$, $Padj = 0.000248$) (Fig. EV1C). The above results suggest that the key gene GLDC may be a potential target involved in the pathogenesis of IgAN.

Since PDGFRβ is expressed in both the glomerular mesangial area and mesangial cells (Buhl et al, 2020; Nakagawa et al, 2011; Neubauer et al, 2013), we used PDGFRβ to characterize mesangial cells in samples from IgAN patients. GLDC/PDGFRβ immunohistochemical double staining was performed in 19 renal samples collected from IgAN patients to determine the expression of GLDC in human glomerular mesangial cells (HMCs). The results of Fig. 1A showed that GLDC could accumulate with PDGFRβ in renal tissues of different Lee stages. To clarify the localization of GLDC in glomeruli, we performed co-staining with CD31 (an endothelial cell marker), podocin (a podocyte marker), and claudin-1 (a parietal epithelial cell marker and a component of crescents) (Fig. EV1D). The result showed that GLDC partially co-localizes with podocin, but not significantly with CD31 or claudin-1. To further compare the co-localization coefficients of GLDC with podocin and PDGFRβ, we used ImageJ software for co-localization analysis. The co-localization coefficient with PDGFRβ was significantly higher than that with podocin (Fig. 1A), indicating that GLDC is primarily localized in glomerular mesangial cells. The correlation analysis between GLDC semi-quantitative analysis results and clinicopathological features of 49 IgAN patients showed that the expression of GLDC was significantly correlated with Lee's grade, segmental glomerulosclerosis (S) and tubular atrophy/interstitial fibrosis (T) (Table 1), indicating that GLDC may have effect on the progression of IgAN.

In addition, the expressions of GLDC and IgA were increased in the IgAN mice compared with the control mice, and PAS staining result showed that mesangial matrix and mesangial cells were increased in the model group (Fig. 1B). In normal mice, the expression of GLDC in glomeruli is very weak, while in the IgAN mouse model, the expression of GLDC is increased and co-localization sites with PDGFRβ are observed (Fig. EV1E). Based on the above results, it is speculated that GLDC may have an effect on the pathological characteristics of HMCs in IgAN.

## GLDC promoted the proliferation of glomerular mesangial cells in vitro

After GLDC overexpression in mouse glomerular mesangial cells (SV40-MES13 cells), the growth rate of SV40-MES13 is significantly faster than that of NC group (Fig. 2A). Compared with control mice, IgAN mice showed more polymerized IgA with molecular weight higher than 180 kDa in both serum and kidney tissues (Fig. EV2A). The IgAN in vitro model was established by stimulating SV40-MES13 cells with polymer IgA (pIgA) extracted from IgAN mouse serum. Then, GLDC interference was performed. The cell viability of SV40-MES13 cells in pIgA stimulation group was increased, and silencing GLDC markedly reduced cell viability (Fig. 2B). Western blot analysis showed that GLDC overexpression in SV40-MES13 cells could positively regulate the protein expressions of PCNA and cyclinD1, which are proliferation factors (Fig. 2C). After pIgA stimulation, the protein expressions of GLDC, PCNA, and cyclinD1 were increased, and further interference with GLDC could reduce these protein expressions (Fig. 2C). In addition, GLDC overexpression decreased G0/G1 phase arrest of SV40-MES-13 cells (Fig. 2D), while interfering with GLDC restored G0/G1 phase arrest of pIgA-stimulated SV40-MES-13 cells (Fig. 2E). The secretion of inflammatory factors (MCP-1 and IL-6), complement activation factors (C3), pro-fibrotic factors (TGF-β1 and TNF-α) were increased in the pIgA stimulation group, while they were decreased after silencing GLDC (Fig. 2F–J).

The effect of GLDC on the chemotactic ability of glomerular mesangial cells to macrophages was investigated. pIgA-stimulated SV40-MES13 cells showed enhanced chemotaxis ability to macrophages, while GLDC silencing reduced the chemotaxis ability to macrophages (Fig. EV2B). In addition, there were no significant changes in the contents of MCP-1, IL-6, C3, TGF-β1, and TNF-α in the supernatant of surviving cells after GLDC silencing (Fig. EV2C), indicating that GLDC did not affect the secretion of cytokines.

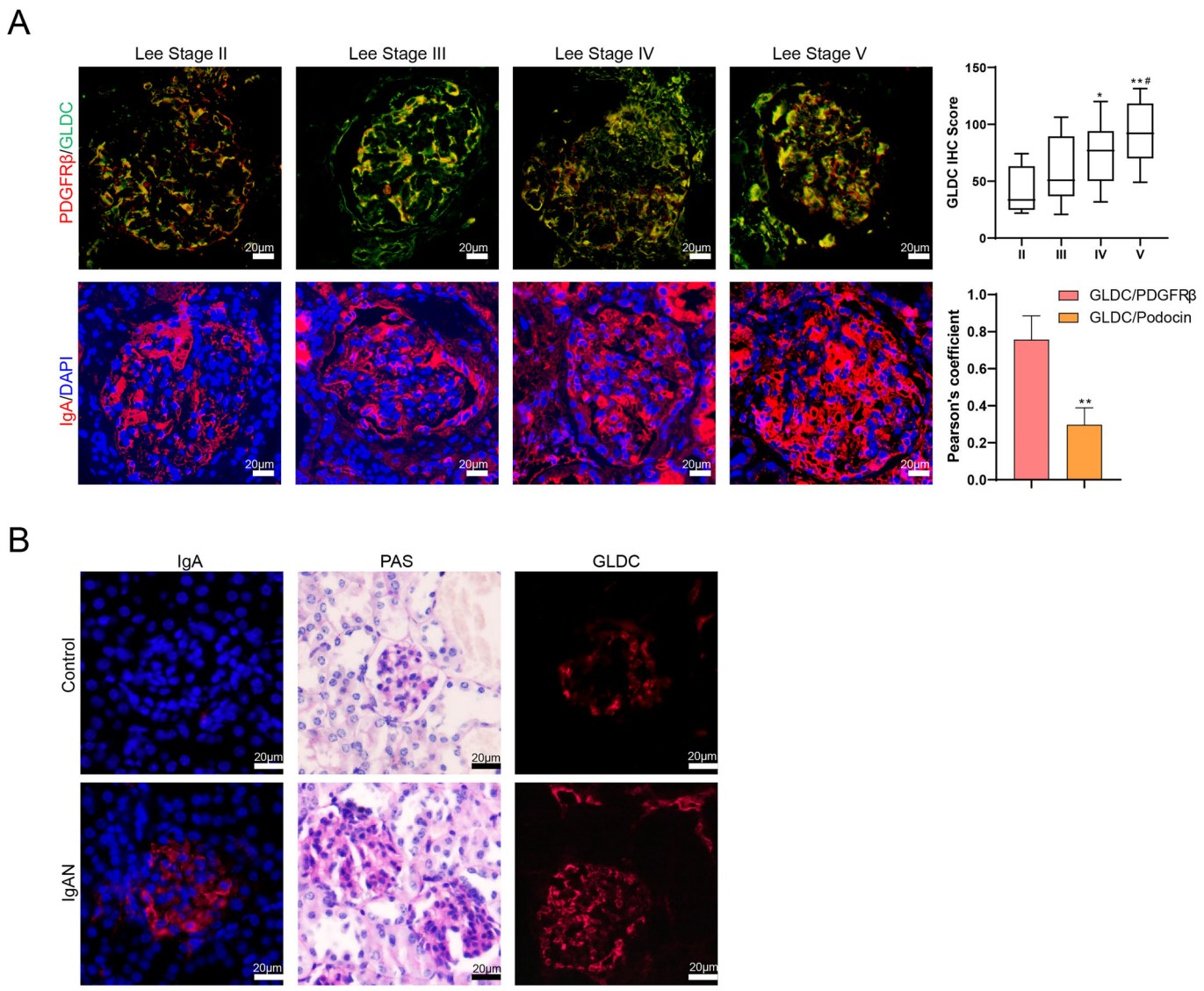

**Figure 1. GLDC was highly expressed in IgAN glomeruli.**

(A) Kidney tissue samples were collected from 49 patients with IgAN. Immunohistochemical staining of IgA, GLDC/PDGFRβ were performed. Scale bar = 20 μm. GLDC IHC Score and Pearson's coefficient were calculated using the ImageJ software. (Upper) One-way ANOVA with Tukey's post-hoc test (Lee II, $N = 13$, Lee III, $N = 13$, Lee IV, $N = 12$, Lee V, $N = 11$). Lee IV vs. Lee II, $*P = 0.0118$; Lee V vs. Lee II, $**P = 0.0002$; Lee V vs. Lee III, $^{\#}P = 0.0463$. (Lower) GLDC/PDGFRβ ($N = 49$), GLDC/Podicin ($N = 29$). Mann–Whitney test, $P < 0.0001$. Error bar: mean ± SD. Three horizontal lines within the box (from top to bottom) represent the 75th percentile, 50th percentile, and 25th percentile of the data, respectively. (B) The expressions of IgA and GLDC in the glomeruli of control mice ($N = 6$) and IgAN mice ($N = 6$) were detected by immunohistochemical staining assay. PAS staining was also performed. Scale bar = 20 μm. Source data are available online for this figure.

## GLDC affected the proliferation of glomerular mesangial cells through an enzyme-dependent manner

Since that energy metabolism is a non-negligible factor in promoting cell proliferation (Zhu and Thompson, 2019), this study then explored whether GLDC promotes glomerular mesangial cell proliferation through a metabolism dependent or independent mechanism. Point mutations were first designed near the catalytic active sites of evolutionarily conserved GLDC to disrupt its metabolic activity. According to previous studies, cofactor pyridoxal phosphate (PLP) binds to GLDC Lys759 (K759), so the amino acid within 5 Å of Lys759 or substrate glycine and

dimerization interface is regarded as the active site residues (Farris et al, 2021). Here, after molecular docking, it was found that in addition to K759, G776 can also bind PLP (Fig. 3A), and the corresponding mutation at the human site is considered to be a nonlethal mutation (Zhang et al, 2012). Therefore, two conserved point mutations mG776R and mK759A corresponding to human hG771R and hK754A were selected (Fig. 3A). Then, GLDC-G776R and GLDC-K759A mutant plasmids were transfected into SV40-MES13 cells to abolish GLDC activity. The results of PCNA protein expression, cell activity, and cell cycle showed that G776R and K759A mutation significantly inhibited glomerular mesangial cell proliferation (Fig. 3B,C). The above results indicated that GLDC

**Table 1. The correlation between the semi-quantitative analysis results of GLDC and the clinicopathological features of IgAN was analyzed (N = 49).**

| Characteristics | | GLDC high | GLDC low | P value |
|---|---|---|---|---|
| Age | ≥43 | 15 | 11 | 0.321 |
| | <43 | 10 | 13 | |
| Sex | Male | 12 | 12 | 1 |
| | Female | 13 | 12 | |
| LEE | II | 3 | 10 | **0.017** |
| | III | 5 | 8 | |
| | IV | 8 | 4 | |
| | V | 9 | 2 | |
| M | M0 | 0 | 1 | 0.302 |
| | M1 | 25 | 23 | |
| E | E0 | 15 | 18 | 0.263 |
| | E1 | 10 | 6 | |
| S | S0 | 5 | 15 | **0.002** |
| | S1 | 20 | 9 | |
| T | T0 | 9 | 18 | **0.019** |
| | T1 | 8 | 4 | |
| | T2 | 8 | 2 | |
| C | C0 | 18 | 17 | 0.31 |
| | C1 | 5 | 7 | |
| | C2 | 2 | 0 | |
| IgA | (2 +) | 12 | 13 | 0.666 |
| | (3 +) | 13 | 11 | |
| C3 | (−) | 5 | 9 | 0.458 |
| | (1 +) | 5 | 4 | |
| | (2 +) | 14 | 11 | |
| | (3 +) | 1 | 0 | |

Note: Mesangial Hypercellularity (M) indicates: M0: Mesangial score ≤0.5. M1: Mesangial score >0.5. Endocapillary Hypercellularity (E) indicates: E0: Absent. E1: Present. Segmental Glomerulosclerosis (S) indicates: S0: Absent. S1: Present. Tubular Atrophy/Interstitial Fibrosis (T) indicates: T0: 0–25%. T1: 26–50%. T2: >50%. Crescent Formation (C) indicates: C0: Absent. C1: Crescents in at least 1 glomerulus and <25% of glomeruli. C2: Crescents in ≥25% of glomeruli. P values in bold indicate statistical significance.

regulated glomerular mesangial cell proliferation with a metabolic enzyme-dependent manner.

## GLDC mediated pyrimidine metabolism to regulate the proliferation of glomerular mesangial cells

Based on the fact that GLDC promotes the proliferation of glomerular mesangial cell cells through a metabolically dependent manner, metabolomic analysis was conducted. LC/MS non-target metabolomics analysis and KEGG analysis were used to determine the key metabolic pathways that GLDC regulates glomerular mesangial cell proliferation, and found that the differentially expressed metabolites of pIgA and pIgA + si-GLDC groups were highly enriched in pyrimidine metabolism, glycine, serine and threonine metabolism, central carbon metabolism, pentose and glucuronate interconversion, bile secretion (Fig. EV3A), the first

four of which are relevant to cell proliferation. Since GLDC is an important enzyme in the glycine cleavage system, key metabolites involved in glycine, serine, and threonine metabolism were decreased in the GLDC knockdown group compared to the pIgA group (Fig. 4A). In addition, pentose and glucuronate interconversions, central metabolism, bile secretion was slightly changed in the pIgA + si-GLDC group compared with pIgA group (Fig. EV3B–D). It is noteworthy that pyrimidine metabolism was inhibited after GLDC knockdown (Fig. 4B). The key metabolites of pyrimidine metabolism, thymidine, thymine and uracil, were increased in SV40-MES13 cells in the wild type (WT) group, accompanied by a downregulation of glycine (Fig. 4C,D). However, the opposite results were observed in GLDC-G776R and GLDC-K759A groups (Fig. 4C,D), suggesting that GLDC contributes to the promotion of pyrimidine metabolism in glomerular mesangial cells in IgAN.

GLDC catalyzes glycine to CH2-THF, a group that cooperates with pyrimidine biosynthesis during cell proliferation (Liu et al, 2021; Zhang et al, 2012). Next, pIgA model group or GLDC overexpression group SV40-MES13 cells were treated with antifolate drug methotrexate to determine whether GLDC regulates pyrimidine metabolism a metabolism-dependent manner. Figure 4E,F shows that the proliferation of SV40-MES13 cells in pIgA model group and GLDC overexpression group was significantly decreased after methotrexate treatment, accompanied by inhibition of pyrimidine metabolism (Fig. 4G,H). These results indicated that the CH2-THF catalyzed by GLDC was the main group involved in regulating pyrimidine metabolism.

In pyrimidine metabolism, CAD and DHODH are two well-known key rate-limiting enzymes (He et al, 2022). In this study, immunohistochemical staining was performed on glomeruli of IgAN model mice, and the results showed that CAD-positive expression was increased (Fig. EV4A). Both CAD and DHODH protein expressions were increased in IgAN in vitro models (Fig. EV4B), suggesting that pyrimidine metabolism may change in IgAN. To further clarify the regulatory role between GLDC and pyrimidine metabolism, we blocked pyrimidine metabolism by silenced CAD or DHODH. We found that compared with GLDC + si-NC group, knockdown of CAD or DHODH resulted in loss of the proliferation promoting effect of GLDC on the glomerular mesangial cells (Fig. 4I). In addition, further silencing of CAD or DHODH did not further downregulate cell growth activity compared to the pIgA + si-GLDC group (Fig. 4J). Our data suggest that conversion of glycine to CH2-THF by GLDC promotes cycling of pyrimidine metabolism, enhancing nucleotide pool, and thus promoting the proliferation of glomerular mesangial cells (Fig. 4K).

## GLDC-regulated pyrimidine synthesis fuels glycolysis in glomerular mesangial cells

Although research suggests that disrupting the downstream metabolic pathways of GLDC—such as purine or cholesterol synthesis—suppresses the expression of cyclins and CDKs, thereby inhibiting cell proliferation (Alptekin et al, 2019), the specific mechanisms by which pyrimidine metabolism mediates the regulation of mesangial cell proliferation by GLDC remain unknown. Interestingly, we also observed that the release trend of lactate is positively correlated with the proliferative capacity of mesangial cells under the regulation of GLDC (Fig. 5A,B). Lactate

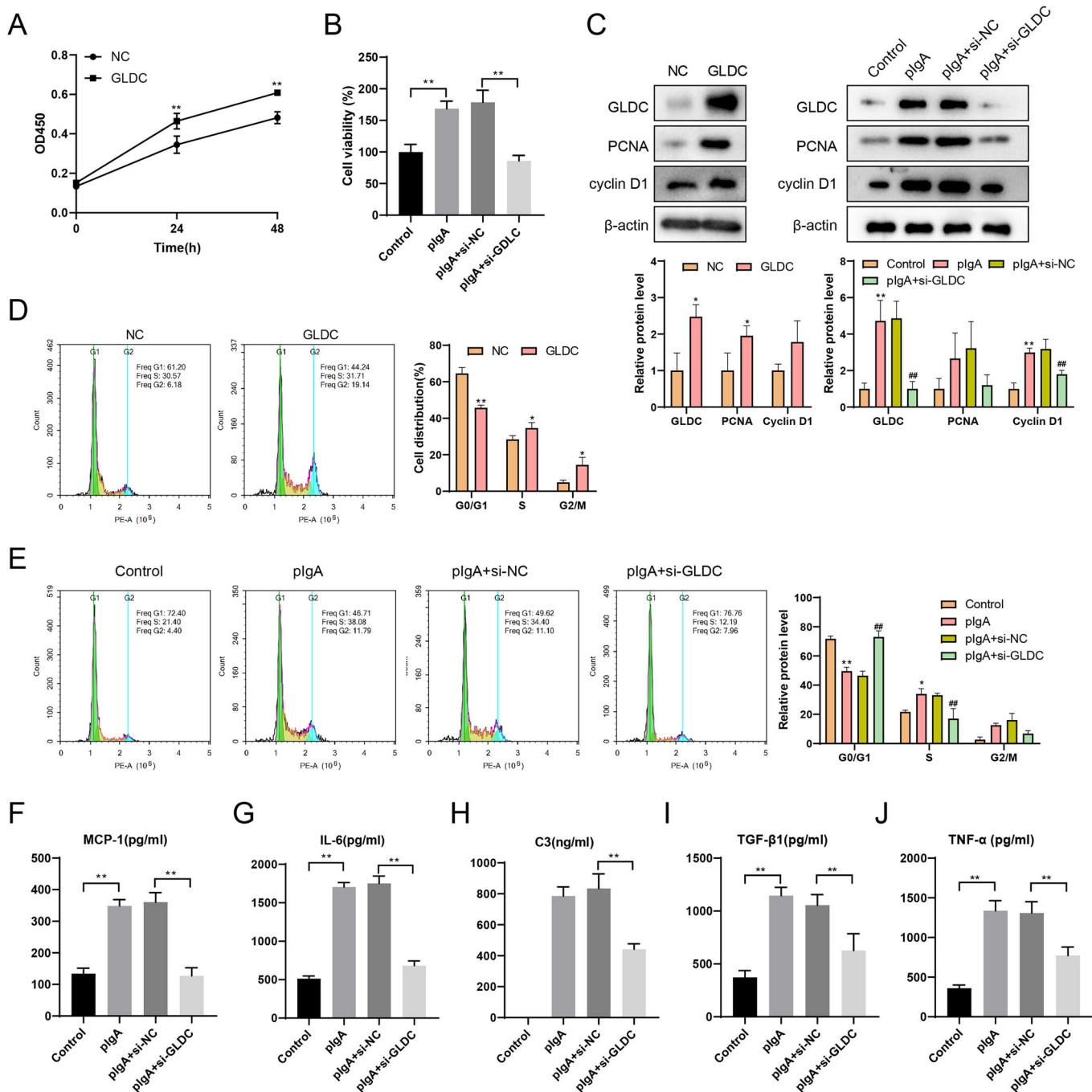

production is a key indicator of glycolysis (Alptekin et al, 2019), and studies have shown that glycolysis plays a role in kidney diseases. For instance, in diabetic nephropathy, glycolysis affects the progression of renal fibrosis (Li et al, 2023), and in lupus nephritis, glycolysis influences the development of nephritis by regulating the proliferation of glomerular mesangial cells (Qi et al, 2023). To confirm the role of the glycolytic pathway in the regulation of mesangial cell growth by GLDC, we treated the GLDC overexpression group with the glycolytic pathway inhibitor 2-DG

(Lee et al, 2022). Under these conditions, GLDC's effects on cell growth (Fig. EV5A), lactate production (Fig. EV5B), and the expression of molecules related to the G1-S transition of the cell cycle (Fig. EV5C,D) were all inhibited. The combination of 2-DG and si-GLDC showed no significant difference in inhibiting pIgA function compared to 2-DG alone, but it was more efficient than si-GLDC treatment alone (Fig. EV5E–G), suggesting that the inhibitory effect of GLDC on mesangial cell proliferation is dependent on its suppression of glycolysis.

**Figure 2. GLDC promotes the proliferation of glomerular mesangial cells in vitro.**

SV40-MES13 cells were grouped into negative control (NC) and GLDC. Mouse GLDC overexpression plasmids were used to overexpress GLDC in SV40-MES13 cells. Additionally, SV40-MES13 cells were grouped into control, pIgA, pIgA + si-NC, pIgA + si-GLDC. Small interfering RNAs (siRNAs) were used to interfere GLDC expression in SV40-MES13 cells. (A, B) Cell viability assay was performed by the CCK-8 assay. (A) Two-way ANOVA with Sidak's post-hoc test, $N = 3$, 24 h GLDC vs. NC, **$P = 0.0025$; 48 h GLDC vs. NC, **$P < 0.0001$. Error bar: mean ± SD. (B) One-way ANOVA with Tukey's post-hoc test, $N = 3$, PIgA vs. Control, **$P = 0.0011$; pIgA+si-NC vs. pIgA+si-GLDC, **$P = 0.0001$. Error bar: mean ± SD. (C) Western blot assay was used to detect GLDC, PCNA, cyclinD1 protein expressions. (Left) Unpaired $T$ test, $N = 3$, GLDC, *$P = 0.0119$; PCNA, *$P = 0.0401$; Cyclin D1, $P = 0.0904$. (Right) One-way ANOVA with Tukey's post-hoc test, $N = 3$, GLDC: pIgA vs. Control, **$P = 0.0016$, pIgA+si-GLDC vs. pIgA+si-NC, ##$P = 0.0013$; Cyclin D1: pIgA vs. Control, **$P = 0.0006$, pIgA+si-GLDC vs. pIgA+si-NC, ##$P = 0.0059$. Error bar: mean ± SD. (D, E) Cell cycle was detected by flow cytometry. Unpaired T test (D), $N = 3$, G0/G1 phase, **$P = 0.0007$; S phase, *$P = 0.0413$; G2/M phase, *$P = 0.0181$. (E) G0/G1 phase: one-way ANOVA with Tukey's post-hoc test, $N = 3$, pIgA vs. Control, **$P < 0.0001$, pIgA+si-GLDC vs. pIgA+si-NC, ##$P < 0.0001$. S phase: one-way ANOVA with Tukey's post-hoc test, $N = 3$, pIgA vs. Control, *$P = 0.0233$, pIgA+si-GLDC vs. pIgA+si-NC, ##$P = 0.005$. G2/M phase: Kruskal–Wallis test with Dunn's post-hoc test, $N = 3$. Error bar: mean ± SD. (F–J) ELISA was used to detect MCP-1, IL-6, C3, TGF-β1, and TNF-α content in supernatants. (F) One-way ANOVA with Tukey's post-hoc test, $N = 3$, pIgA vs. Control, **$P < 0.0001$, pIgA+si-GLDC vs. pIgA+si-NC, **$P < 0.0001$. (G) One-way ANOVA with Tukey's post-hoc test, $N = 3$, pIgA vs. Control, **$P < 0.0001$, pIgA+si-GLDC vs. pIgA+si-NC, **$P < 0.0001$. (H) One-way ANOVA with Tukey's post-hoc test, $N = 3$, pIgA+si-GLDC vs. pIgA+si-NC, **$P = 0.0009$. (I) One-way ANOVA with Tukey's post-hoc test, $N = 3$, pIgA vs. Control, **$P < 0.0001$, pIgA+si-GLDC vs. pIgA+si-NC, **$P = 0.0051$. (J) One-way ANOVA with Tukey's post-hoc test, $N = 3$, pIgA vs. Control, **$P < 0.0001$, pIgA+si-GLDC vs. pIgA+si-NC, **$P < 0.0016$. Error bar: mean ± SD. Source data are available online for this figure.

Studies have suggested that pyrimidine metabolism influences the occurrence of glycolysis (He et al, 2022; Zhou et al, 2023). Pyrimidine synthesis plays a key role in glycolytic metabolism and functions as a metabolic vulnerability (He et al, 2022). Consistent with these findings, we observed that when we knocked down the rate-limiting enzyme of pyrimidine metabolism, the lactate secretion-promoting ability of GLDC was downregulated (Fig. 5C,D). This leads us to hypothesize that glycolysis is modulated during the combined treatment of GLDC and the rate-limiting enzyme of pyrimidine metabolism.

To further confirm whether GLDC regulates glycolysis through pyrimidine metabolism, we examined the mRNA levels of key glycolytic pathway molecules, including SLC2A1, HK2, PGK1, PKM, and LDHA. Under GLDC overexpression conditions, the expression levels of these molecules were all increased. However, this promoting effect of GLDC on the aforementioned molecules was reversed upon knockdown of CAD or DHODH (Fig. 5E). Moreover, pIgA could not promote the expression of these molecules in cells with silenced GLDC (Fig. 5F). Additionally, in HMCs, we found that GLDC promotes extracellular acidification rate (ECAR), which is a measure of glycolytic activity. This effect of GLDC was inhibited upon knockdown of CAD or DHODH (Fig. 5G,H). Therefore, GLDC regulates glycolysis through pyrimidine metabolism and affects the transcriptional expression levels of key glycolytic molecules.

## Silencing GLDC in vivo alleviated IgAN progression

Next, an in vivo model was used to verify the effect of interfering with GLDC on IgAN progression. After injection of AAV-sh-GLDC into renal cortex, there was little significant change in IgA deposition, with a slight downregulation, while there was a significant decrease in C3, and PAS positive areas (mesangial matrix and mesangial cells) were also decreased (Fig. 6A), indicating that GLDC knockdown cannot inhibit IgA formation, but it can inhibit the accumulation of C3 in IgAN processes.

Mononuclear/macrophage infiltrates into the kidney tissues and exacerbates kidney injury by secreting inflammatory cytokines, which are related to the formation of glomerular crescent (Urushihara et al, 2020). Therefore, we further investigated the positive expression of monocyte/macrophage marker CD68. The result showed that after silencing GLDC in vivo, the GLDC8-positive cells and CD68-positive

cells were reduced (Fig. 6B), indicating that monocyte/macrophage infiltration was reduced. This may be related to the decrease of MCP-1 and IL-6 protein expressions after interfering with GLDC (Fig. 6C). This result is consistent with the analysis of the correlation between GLDC-containing modules and crescent formation in the clinical IgAN dataset (Fig. EV1).

## Silencing GLDC in vivo suppressed pyrimidine metabolism and glycolysis in glomeruli

The effects of GLDC silence on pyrimidine metabolism and glycolysis were next investigated in vivo. Mice were grouped into control, IgAN, IgAN + AAV-sh-NC, IgAN + AAV-sh-GLDC groups. Glomeruli were isolated from the above groups, and non-target metabolomics analysis and KEGG pathway enrichment analysis were performed. Figure 7A shows that the pyrimidine metabolism was changed in IgA group as well as AAV-sh-GLDC group, and related metabolites (glutamine, thymidine, uracil) were highly expressed in the IgA group and reduced after GLDC silencing (Fig. 7B). In addition, GLDC silencing in vivo significantly reduced the production of lactate, accompanied by an increase in glycine (Fig. 7C,D), indicating that glycolysis process was inhibited. The above results are consistent with the results of in vitro assays, suggesting that GLDC promotes the growth of glomerular mesangial cells in IgAN by mediating pyrimidine metabolism and glycolytic pathways.

# Discussion

IgAN, one of the most common types of primary glomerular disease, often occurring in children and young adults, and the onset of the disease is usually insidious, with diverse clinical manifestations (Stamellou et al, 2023). The typical pathological features of IgAN are the presence of immune complexes dominated by IgA or IgA1 (especially Gd-IgA1) in the glomerular mesangial region, with or without deposition of other immunoglobulins, accompanied by mesangial expansion, glomerular mesangial cell proliferation and mesangial matrix production (Jash et al, 2024). The pathogenesis of IgAN has not been fully elucidated, and there is a lack of targeted clinical treatment. Thus, it is very important to further elucidate IgAN pathogenesis and seek new, effective and specific treatment. This study is based on bioinformatics to search for IgAN-related

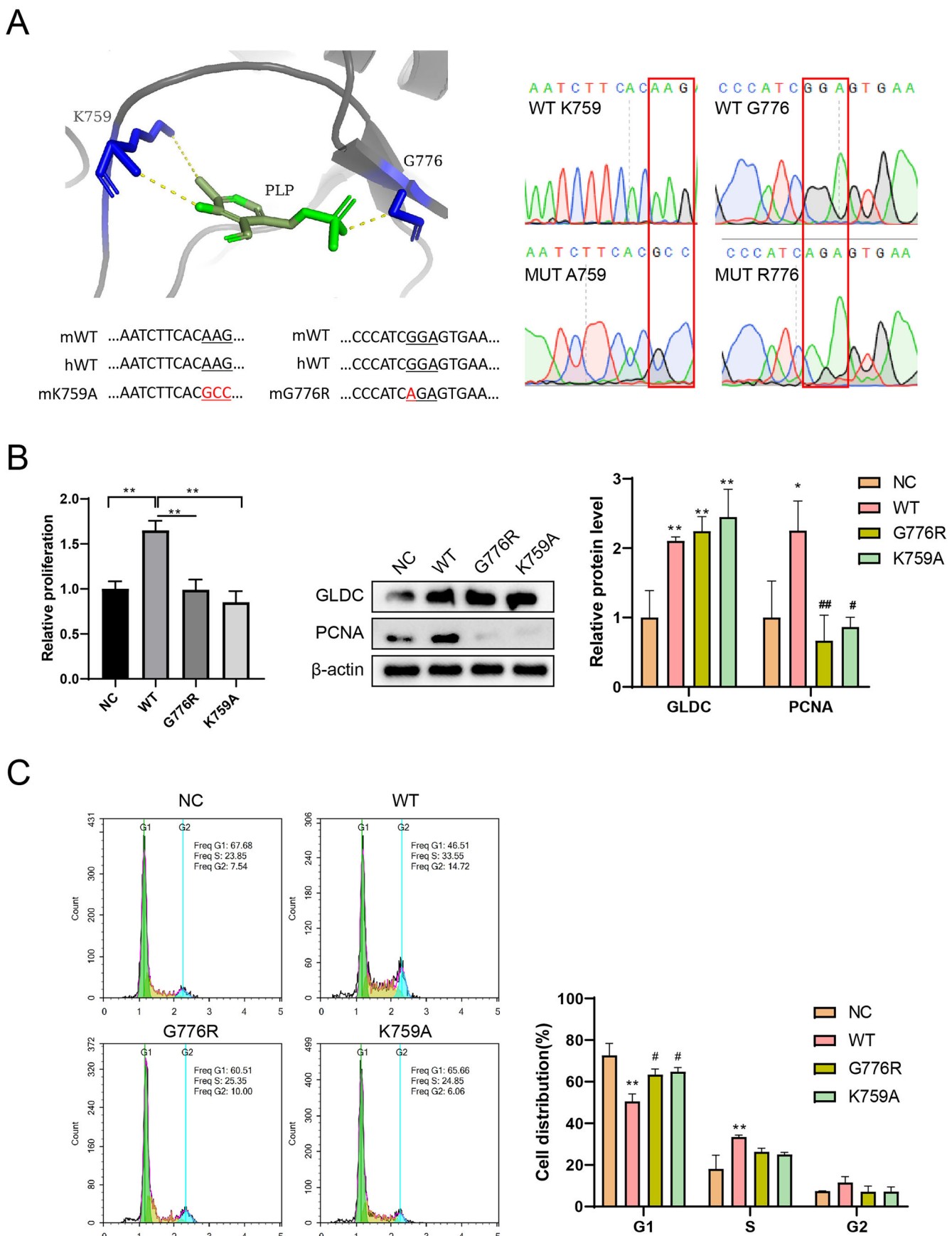

**Figure 3. GLDC affected the proliferation of glomerular mesangial cells through an enzyme-dependent manner.**

(A) K759 and G776 were selected for point mutation. (B, C) GLDC-G776R or GLDC-K759A mutant plasmid was transfected into SV40-MES13 cells, and grouped into NC, GLDC-WT (WT), GLDC-G776R (G776R), GLDC-K759A (K759A). (B) Cell proliferation assay was performed. Western blot assay was used to detect GLDC and PCNA protein expressions. (Left) One-way ANOVA with Tukey's post-hoc test, $N = 3$, NC $vs.$ WT, $**P = 0.0004$; WT $vs.$ G776R, $**P = 0.0003$; WT $vs.$ K759A, $**P < 0.0001$. (Right) One-way ANOVA with Tukey's post-hoc, $N = 3$, GLDC: NC $vs.$ WT, $**P = 0.0084$; NC $vs.$ G776R, $**P = 0.0041$; NC $vs.$ K759A, $**P = 0.0016$. PCNA: NC $vs.$ WT, $*P = 0.0185$; WT $vs.$ G776R, $##P = 0.0048$; WT $vs.$ K759A, $#P = 0.0105$. Error bar: mean ± SD. (C) Cell cycle was detected by flow cytometry. G0/G1 phase: One-way ANOVA with Tukey's post-hoc test, $N = 3$, NC $vs.$ WT, $**P = 0.0005$; WT $vs.$ G776R, $#P = 0.0144$; WT $vs.$ K759A, $#P = 0.0079$. S phase: one-way ANOVA with Tukey's post-hoc test, $N = 3$, NC $vs.$ WT, $**P = 0.0029$. G2/M phase: One-way ANOVA with Tukey's post-hoc test, $N = 3$. Error bar: mean ± SD. Source data are available online for this figure.

biomarkers in order to provide a new direction for the early diagnosis and treatment of IgAN, and found that high expression of GLDC was correlated with the increase in Lee grading (Fig. 8). This is also the first study to examine the effects of GLDC on the development of IgAN.

Glycine degradation occurs mainly through the GCS, which is mediated by the oxidoreductase GLDC (Weaver et al, 2022). Many studies have found that amino acid metabolic enzymes are closely related to biological behaviors such as tumor cell proliferation, including GLDC. In this study, GLDC interference in pIgA stimulated in vitro model reduced the activity and glomerular mesangial cell proliferation, and the expression of inflammatory molecules and complement activation-related molecules was also decreased. This is the first study to confirm the new biological function of GLDC in promoting the proliferation of glomerular mesangial cells in IgAN. What is the molecular mechanism of GLDC-mediated mesangial cell proliferation in IgAN? Based on the fact that cellular metabolism is the basis of all biological activities (Zhu and Thompson, 2019) and GLDC is a metabolic enzyme, this study first explored whether GLDC promotes mesangial cell proliferation through a metabolically dependent or independent mechanism. After mutation of two active sites, neither GLDC-G776R nor GLDC-K759A significantly promoted the proliferation of mesangial cell. This reminds us that GLDC regulates the growth of MC dependent on its metabolism activity. It was found that GLDC was overexpressed in non-small cell lung cancer stem cells and could enhance the tumorigenic ability of non-small cell lung cancer stem cells by enhancing glycolysis and pyrimidine metabolism (Zhang et al, 2012). At present, the pathway of pyrimidine metabolic degradation in microorganisms has been deeply studied, but there are relatively few studies on pyrimidine metabolism in other species. Here, pyrimidine metabolism was significantly inhibited after GLDC knockdown in IgAN in vitro model, and the ability of GLDC to promote cell proliferation is lost after blocking pyrimidine metabolism by silenced CAD or DHODH treatment. The completion of our study will contribute to a better understanding of metabolism in the human body.

In some renal diseases, the renal tissues showed high glucose uptake rate, increased expression of key enzymes of glycolysis reaction, and enhanced glycolysis with increased lactic acid as the end product of metabolism, suggesting that aerobic glycolysis plays an important role in the renal diseases' development (Lian et al, 2019; Shen et al, 2020; Simon et al, 2020). Qing et al explored the metabolic function changes of renal intrinsic cells in IgAN patients, in which glycolysis in podocytes was significantly increased, and glycolysis in proximal tubular cells was decreased, while metabolic reprogramming in glomerular mesangial cell was not significantly

changed (Qing et al, 2024), but this was not in conflict with the results of this study. In this study, we found that overexpression of GLDC promoted glycolysis of normal mesangial cells, and further blocking pyrimidine metabolism can inhibit the glycolysis. In vivo experiments further confirmed that GLDC silencing could inhibit pyrimidine metabolism and glycolysis, thus alleviating IgAN.

There are still shortcomings in this study. First, the number of clinical samples is small. It is necessary to continue to collect clinical data of IgAN patients to further analyze the clinical significance of GLDC. Second, how IgA regulates the expression of GLDC in glomerular mesangial cells remains to be further explored. In IgAN disease, IgA is overproduced and aggregates with IgA autoantibodies to form immune complex, and IgA immune complexes carrying anti-βII-spectrin are more likely to bind to glomerular mesangial cell surface, leading to IgA deposition in glomeruli and pathological changes in glomerular mesangial cells (Nihei et al, 2023). However, it has also been reported that aggregated IgA can directly regulate mesangial cell proliferation (Xia et al, 2020; Zhu et al, 2022). Therefore, in the future, we can further explore whether aggregated IgA or IgA immune complex with autoantibodies induces GLDC-related proliferation changes in glomerular mesangial cells. Then, due to the limited source of HMCs, most in vitro studies used murine mesangial cell lines, which need to be further verified on human cell line samples.

In conclusion, GLDC plays an important role in regulating the proliferation of glomerular mesangial cells through pyrimidine metabolism and glycolysis, and this regulatory mechanism provides an important theoretical basis for developing new targets for IgAN.

# Methods

**Reagents and tools table**

| Reagent/resource | Reference or Source | Catalog Number |
|---|---|---|
| **Experimental models** | | |
| SV40-MES13 (*M. musculus*) | Procell Life Science & Technology | CL-0470 |
| Primary human glomerular mesangial cell (*H. sapiens*) | Procell Life Science & Technology | CP-H067 |
| BALB/c (*M. musculus*) | Changzhou Cavens Laboratory Animal Co. Ltd. | N/A |
| IgAN Patient kidney tissue biopsy sample (*H. sapiens*) | Second Affiliated Hospital, Jiangxi Medical College, Nanchang University | N/A |

| Reagent/resource | Reference or Source | Catalog Number |
|---|---|---|
| **Recombinant DNA** | | |
| pcDNA3.1(+)-GLDC-eGFP (M.musculus) | GeneCreate | N/A |
| pcDNA3.1(+)-G776R GLDC-eGFP (M. musculus) | GeneCreate | N/A |
| pcDNA3.1(+)-K759A GLDC-eGFP (M. musculus) | GeneCreate | N/A |
| pcDNA3.1(+)-GLDC-eGFP (H. sapiens) | GeneCreate | N/A |
| AAV-sh-NC | GeneChem | N/A |
| AAV-sh-GLDC (M. musculus) | GeneChem | N/A |
| **Antibodies** | | |
| Rabbit anti-GLDC | Proteintech | 24827-1-AP |
| Mouse anti-GLDC | Invitrogen | MA5-49279 |
| Rabbit anti-IgA | Proteintech | 11449-1-AP |
| Rabbit anti-C3 | Abcam | ab200999 |
| Rabbit anti-CD68 | Abcam | ab283654 |
| Mouse anti-PDGFRβ | Santa cruz | sc-374573 |
| Rabbit anti-PDGFRβ | Abcam | ab313777 |
| Rabbit anti-CAD | Abcam | ab99312 |
| Mouse anti-CD31 | Abcam | ab9498 |
| Rabbit anti-Podocin | Abcam | ab229037 |
| Rabbit anti-Claudin-1 | Abcam | ab180158 |
| Mouse anti-PCNA | Santa Cruz | sc-56 |
| Rabbit anti-CyclinD1 | Abcam | ab134175 |
| Rabbit anti-DHODH | Abcam | ab174288 |
| Mouse anti-β-Actin | Beyotime Biotechnology | AF2811 |
| **Oligonucleotides and other sequence-based reagents** | | |
| siRNA sequence | This study | Table 2 |
| PCR primers | This study | Table 3 |
| **Chemicals, enzymes and other reagents** | | |
| Bovine serum albumin (BSA) | Sigma | V900933 |
| lipopolysaccharide (LPS) | Sigma | L2630 |
| carbon tetrachloride (CCl$_4$) | MCE | HY-Y0298 |
| Castor oil | Macklin | C15754589 |
| DMEM/F12 | Procell Life Science & Technology | PM150310 |
| Fetal bovine serum (FBS) | Yeasen Biotechnology (Shanghai) Co., Ltd. | 40130ES50 |
| Penicillin/streptomycin | Hyclone | SV30010 |
| Lipofectamine 3000 | Invitrogen | |
| 2-Deoxy-D-glucose (2-DG) | MCE | HY-13966 |
| Methotrexate | MCE | HY-14519 |

| Reagent/resource | Reference or Source | Catalog Number |
|---|---|---|
| Propyl iodide | ThermoFisher | R37169 |
| Glucose | Procell Life Science & Technology | PB180418 |
| Oligomycin | MCE | HY-N6782 |
| Mitochondrial uncoupler | MCE | HY-100410 |
| Rotenone | MCE | HY-B1756 |
| Antimycin A | Santa Cruz | sc-202467 |
| **Software** | | |
| GraphPad Prism version 8.0.2(263) | https://www.graphpad.com/ | N/A |
| SPSS version 19 | https://www.ibm.com/products/spss-statistics | N/A |
| R version 4.5.1 | https://www.r-project.org/ | N/A |
| AutoDock-Vina version 1.2.7 | https://github.com/ccsb-scripps/AutoDock-Vina/releases | N/A |
| PyMOL version 3.1.6 | https://www.pymol.org/ | N/A |
| ImageJ version 1.53C | https://imagej.net/ij/ | N/A |
| SnapGene version 6.0.2 | https://www.snapgene.com/ | N/A |
| **Other** | | |
| Enzyme-labeled instrument | ThermoFisher | Multiskan FC |
| Liquid Chromatography Instrument | ThermoFisher | Vanquish |
| Mass spectrometer | ThermoFisher | Q Exactive |
| Flow Cytometer | Beckman Coulter Life Sciences | CytoFLEX S |
| Microscope | Olympus | IX73 |
| Extracellular Flux Analyzer | Agilent Technologies | XFe24 |
| Real-Time PCR instrument | Applied Biosystems | 7300 |
| Periodic Acid Schiff(PAS) Stain Kit | Solarbio | G1281 |
| Mouse MCP-1 ELISA Kit | Beyotime Biotechnology | PC125 |
| Mouse IL-6 ELISA Kit | Beyotime Biotechnology | PI326 |
| Mouse C3(Complement Component 3) ELISA Kit | Elabscience | E-EL-M0330 |
| TGF-β1(Transforming Growth Factor Beta 1) ELISA Kit | Elabscience | E-EL-0162 |
| Mouse TNF-α(Tumor Necrosis Factor Alpha) ELISA Kit | Elabscience | E-EL-M3063 |
| Cell Counting Kit-8 | Beyotime Biotechnology | C0037 |
| ELISA Kit for Glycine (Gly) | CLOUD-CLONE CORP. WUHAN | CES117Ge |
| lactic acid detection kit | Solarbio | BC2235 |
| Seahorse XF Glycolytic Rate Assay Kit | Agilent | 103344-100 |
| Seahorse XFp Cell Mito Stress Test Kit | Agilent | 103015-100 |
| HiScript III RT SuperMix reagent | Vazyme Biotech Co., Ltd. | R323-01 |
| SYBR qPCR Master Mix | Vazyme Biotech Co., Ltd. | Q711-02 |

## Clinical information

Forty-nine patients with primary IgAN admitted to the Second Affiliated Hospital, Jiangxi Medical College, Nanchang University from December 2023 to December 2024 were recruited. Among them, 8 were males and 11 were females; 3 were Lee stage II, 13 were Lee stage III, and 3 were Lee stage IV. The Oxford Classification of IgAN is used to help predict the risk of disease progression in individual patients (Howie and Lalayiannis, 2023). It includes the following five pathological features: Mesangial Hypercellularity (M): M0: Mesangial score ≤0.5. M1: Mesangial score >0.5; Endocapillary Hypercellularity (E): E0: Absent. E1: Present; Segmental Glomerulosclerosis (S): S0: Absent. S1: Present; Tubular Atrophy/Interstitial Fibrosis (T): T0: 0–25%. T1: 26%–50%. T2: >50%; Crescent Formation (C): C0: Absent. C1: Crescents in at least 1 glomerulus and <25% of glomeruli. C2: Crescents in ≥25% of glomeruli. The clinical information of recruited patients is shown in Table 1. Percutaneous renal biopsy was used to extract kidney tissues for follow-up study.

This study was approved by the Medical Ethics Committee of the Second Affiliated Hospital, Jiangxi Medical College, Nanchang University, and all IgAN patients signed informed consent. Informed consent was obtained from all human subjects and confirm that the experiments conformed to the principles set out in the WMA Declaration of Helsinki and the Department of Health and Human Services Belmont Report.

## Bioinformatics analysis

The GSE141295 dataset with 14 IgAN samples and 10 normal control samples was downloaded from the GEO database (Park et al, 2020) Data ref: (Park et al, 2020). The R package DESeq2 was used for differential expression analysis of the expression matrix (IgAN vs Normal). Adjusted $P$ value ($P$adj) <0.05 was used to screen differentially expressed (DE) genes, and log2 Fold Change (logFC) > 0 was considered to be significantly upregulated. logFC <0 was considered to be significantly downregulated. A total of 4154 DE genes were obtained, of which 2439 were significantly upregulated and 1715 were significantly downregulated. Weighted correlation network analysis (WGCNA) identified key modules in which DE genes expression profiles were significantly correlated with clinical traits in 14 IgAN samples.

## Chemotaxis experiment of Raw264.7 macrophages

The chemotactic ability of SV40-MES13 cells to Raw264.7 macrophages was evaluated in transwell (Corning) chemotaxis assay. The inserts of transwell with 8-mm pore size were pre-treated with 1% BSA for 1 h, and Raw264.7 macrophages were inoculated into the inserts. Raw264.7 macrophages were co-cultured with pIgA or pIgA + si-GLDC group SV40-MES13 cells in the lower chamber for 24 h. After washed with PBS and fixed with methanol for 20 min, the cells were properly air-dried, and stained with 0.1% crystal violet solution (Beijing Solarbio Science & Technology Co., Ltd.) for 30 min. Three fields were selected to count the number of macrophages in the lower chamber under the optical microscope.

## Construction of IgAN mouse model and grouping

BALB/c mice (weight 20–25 g) obtained from Changzhou Cavens Laboratory Animal Co. Ltd. was used for in vivo modeling. IgAN model mice was established by oral administration of bovine serum albumin (BSA), and injection of lipopolysaccharide (LPS) and carbon tetrachloride (CCl₄). Specifically, IgAN mice were gavaged with 800 mg/kg BSA (Sigma) acidified water once every other day. CCl₄ solution (CCl₄: castor oil = 1:5, Sigma) was injected subcutaneously once a week, 0.1 mL/mouse and injected intraperitoneally every 2 weeks, 0.06–0.08 mL/mouse. At week 6 and week 8, 50 μg LPS (Sigma) was injected into the tail vein. The normal control group was given equal amount of acidified water, castor oil and normal saline by gavage, subcutaneous injection and tail vein injection, respectively. At the end of the 11th week, the success of modeling was confirmed by IgA immunohistochemical staining. Glomeruli, kidney tissues and serum were collected for subsequent experiments. This study was approved by the Animal Experimental Ethics Committee of the Second Affiliated Hospital, Jiangxi Medical College, Nanchang University.

In the adreno-associated virus AAV-sh-NC or AAV-sh-GLDC group, after 6 weeks of modeling, kidney orthotopic injection of AAV-sh-NC or AAV-sh-GLDC ($1 \times 10^9$ IU/μL) was performed. After anesthesia, the left renal pedicle was temporarily blocked and 50 μL AAV mixture was injected into the renal cortex with 31 G needle to induce GLDC interference in the animal model.

## Immunohistochemical staining

The positive expressions of IgA and GLDC/platelet-derived growth factor receptor β (PDGFRβ) in clinical kidney tissue samples, and the positive expressions of IgA, GLDC, C3, CD68, CAD in glomeruli of mice were detected by immunohistochemistry staining (Mrohs et al, 2023). The paraffin sections (4 μm) were dewaxed and hydrated, repaired with microwave, and washed with PBS (pH 7.4) after cooling. 3% hydrogen peroxide solution was used to block endogenous peroxidase, and then incubated at 37 °C for 15 min, rinsed with PBS, added primary antibodies with appropriate concentration, respectively. The sections were rinsed with PBS, labeled with horseradish peroxidase, then DAPI color solution was added, and then observed with a fluorescence microscope (Olympus) after restaining and sealing. Antibodies information used are as follows: anti-IgA (Proteintech, 1:200), anti-PDGFRβ (Santa and Abcam, 1:50 for Santa Cruz and 1:100 for Abcam), anti-GLDC (Proteintech and Invitrogen, 1:200 for Proteintech and 1:200 for Invitrogen), anti-C3 (Abcam, 1:500), anti-CD68 (Abcam, 1:100), anti-CAD (Abcam, 1:500), CD31 (Abcam, 1:2000), Podocin (Abcam, 1:500), Claudin-1 (Abcam, 1:100). Clinical sample GLDC expressions intensity was evaluated by semi-quantitative scores, which were 0 (negative), 1 (weak), 2 (moderate), and 3 (strong), respectively, and total IHC scores of GLDC in clinical samples were obtained by multiplying the intensity score with the percentage of positive staining.

## Periodic acid-Schiff (PAS) staining

Kidney tissue samples of mice were collected and fixed with 4% paraformaldehyde, then sliced (4 μm) after paraffin embedding and PAS staining, observed under an optical microscope and photographed.

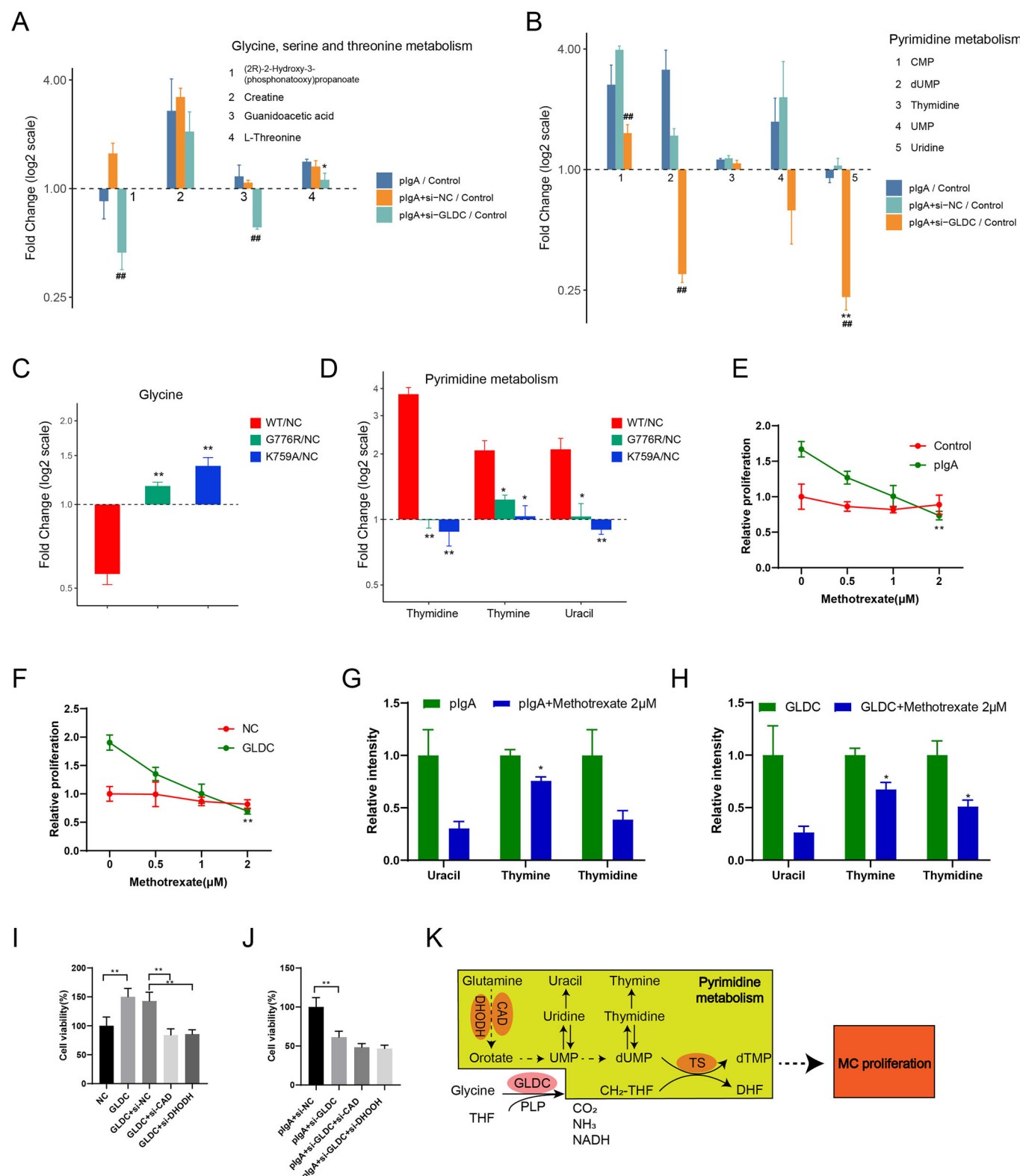

## Identification and purification of pIgA

The protein expressions of IgA in the serum and kidney tissue of control and IgAN group mice were detected by western blot assay. Next, the pooled IgAN murine serum was centrifuged and the supernatant was treated with sodium sulfate to precipitate the protein. The obtained protein components were purified by affinity chromatography to obtain purified IgA, and the possible residual IgG was removed.

**Figure 4. GLDC mediated pyrimidine metabolism to regulate the proliferation of glomerular mesangial cells.**

(A, B) The metabolites of glycine, serine and threonine metabolism, pyrimidine metabolism in the pIgA + si-NC and pIgA + si-GLDC groups were analyzed. (A) Unpaired t test, $N = 3$, No. 1: pIgA+si-GLDC/Control vs. pIgA+si-NC/Control, $^{##}P = 0.0078$. No. 3: pIgA+si-GLDC/Control vs. pIgA+si-NC/Control, $^{##}P = 0.0013$. No. 4: pIgA+si-GLDC/Control vs. pIgA/Control, $^{*}P = 0.0482$. Error bar: mean ± SEM. (B) Unpaired t test, $N = 3$, No. 1: pIgA+si-GLDC/Control vs. pIgA+si-NC/Control, $^{##}P = 0.0075$. No. 2: pIgA+si-GLDC/Control vs. pIgA+si-NC/Control, $^{##}P = 0.0006$. No. 5: pIgA+si-GLDC/Control vs. pIgA/Control, $^{**}P = 0.0002$; pIgA+si-GLDC/Control vs. pIgA+si-NC/Control, $^{##}P = 0.0100$. Error bar: mean ± SEM. (C, D) Glycine, thymidine, thymine, uracil levels in the WT, G776R, K759A groups were detected by ELISA. (C) Unpaired t test, $N = 3$, G776R/NC vs. WT/NC, $^{**}P = 0.0006$; K759A/NC vs. WT/NC, $^{**}P = 0.0017$. Error bar: mean ± SEM. (D) Unpaired t test, $N = 3$, Thymidine: G776R/NC vs. WT/NC, $^{**}P = 0.0005$; K759A/NC vs. WT/NC, $^{**}P = 0.0006$. Thymine: G776R/NC vs. WT/NC, $^{*}P = 0.0200$; K759A/NC vs. WT/NC, $^{*}P = 0.0141$. Uracil: G776R/NC vs. WT/NC, $^{*}P = 0.0162$; K759A/NC vs. WT/NC, $^{**}P = 0.0067$. Error bar: mean ± SEM. (E, F) SV40-MES13 cells treated with methotrexate (0.5, 1.0, 2.0 μM) were grouped into control and pIgA groups, NC and GLDC groups. Cell proliferation assay was performed. (E) One-way ANOVA with Tukey's post-hoc test, $N = 3$, pIgA: Methotrexate 2 μM vs. Methotrexate 0 μM, $^{**}P < 0.0001$. (F) One-way ANOVA with Tukey's post-hoc test, $N = 3$. GLDC: Methotrexate 2 μM vs. Methotrexate 0 μM, $^{**}P < 0.0001$. Error bar: mean ± SD. (G, H) Thymidine, thymine, uracil levels in the pIgA, pIgA + Methotrexate, GLDC, GLDC + Methotrexate groups were detected by ELISA. (G) Unpaired T test, $N = 3$, Thymine, $^{*}P = 0.0228$. (H) Unpaired t test, $N = 3$, Thymine, $^{*}P = 0.0254$; Thymidine, $^{*}P = 0.0303$. Error bar: mean ± SEM. (I, J) SV40-MES13 cells were grouped into NC, GLDC, GLDC + si-CAD, GLDC + si-DHODH, pIgA + si-NC, pIgA + si-GLDC, pIgA + si-GLDC + si-CAD, pIgA + si-GLDC + si-DHODH. Cell viability assay was performed. (I) One-way ANOVA with Tukey's post-hoc test, $N = 3$, NC vs. GLDC, $^{**}P = 0.0059$; GLDC+si-NC vs. GLDC+si-CAD, $^{**}P = 0.0018$; GLDC+si-NC vs. GLDC+si-DHODH, $^{**}P = 0.0022$. (J) One-way ANOVA with Tukey's post-hoc test, $N = 3$, pIgA+si-NC vs. pIgA+si-GLDC, $^{**}P = 0.0014$. Error bar: mean ± SD. (K) The mechanism diagram of GLDC promoting pyrimidine metabolism leading to increased proliferation of glomerular mesangial cells under pIgA treatment. Source data are available online for this figure.

## Cell culture and treatment

Mouse glomerular mesangial cell line SV40-MES13 and human glomerular mesangial cells (HMCs) (Procell Life Science & Technology) were cultured in the DMEM/F12 medium containing 5% fetal bovine serum [FBS, Yeasen Biotechnology (Shanghai) Co., Ltd.] and 1% penicillin/streptomycin (Hyclone).

SV40-MES13 or HMCs were transfected with the plasmids or siRNAs according to Lipofectamine 3000 transfection kit (Invitrogen). The vectors constructed using mouse GLDC are as follows: pcDNA3.1(+)-GLDC-eGFP, pcDNA3.1(+)-G776R GLDC-eGFP, and pcDNA3.1(+)-K759A GLDC-eGFP. Additionally, the vector constructed using human GLDC is named pcDNA3.1(+)-GLDC-eGFP. The control plasmid used is pcDNA3.1(+)-eGFP (provided by Wuhan GeneCreate).

In the GLDC overexpression group, SV40-MES13 cells were transfected with a plasmid carrying the mouse GLDC gene. In the pIgA + si-GLDC/si-NC group, SV40-MES13 cells transfected with GLDC small interfering RNA (si-GLDC, Shanghai GeneChem) or si-NC was cultured in serum-free culture medium for 12 h, and then incubated with 25 μg/mL pIgA for another 48 h. In the GLDC-G776R (G776R) or GLDC-K759A (K759A) group, SV40-MES13 cells were transfected with pcDNA3.1(+)-G776R GLDC-eGFP or pcDNA3.1(+)-K759A GLDC-eGFP. In the pIgA + Methotrexate group, SV40-MES13 cells were cultured with 25 μg/mL pIgA and methotrexate (0.5, 1.0, 2.0 μM). In the GLDC + Methotrexate group, SV40-MES13 cells were transfected with mouse GLDC overexpression plasmid and treated with methotrexate (0.5, 1.0, 2.0 μM). In the GLDC + si-CAD/si-DHODH group, SV40-MES13 cells or HMCs were transfected with mouse/human GLDC overexpression plasmid and si-CAD or si-DHODH. In the pIgA + si-GLDC + si-CAD/si-DHODH group, SV40-MES13 cells or HMCs were transfected si-GLDC + si-CAD/si-DHODH and treated with 25 μg/mL pIgA. In the GLDC + 2-deoxy-D-glucose (2-DG) group, SV40-MES13 cells were transfected with mouse GLDC overexpression plasmid and treated with 10 mM 2-DG.

The transfection details are as follows: pcDNA vectors are transfected into the culture plates at a concentration of 1 μg/ml with pcDNA3.1(+)-eGFP used as the negative control. si-RNA is transfected at a concentration of 50 nM with si-NC serving as the negative control. After 24 h of transfection, SV40-MES cells are cultured in serum-free medium for 12 h and this time point is designated as the starting point for subsequent treatments if pIgA treatment is required. pIgA is added at a concentration of 25 μg/ml at this time point and the cells are further treated for 48 h after which supernatants or cells are collected for downstream assays. If pIgA treatment is not required, the 12-h post-transfection time point is considered the starting point for treatment and downstream assays are conducted after an additional 48 h transfection efficiency is evaluated by qPCR or western blot analysis of the target gene. Drug treatments such as methotrexate are also initiated at the starting point and are synchronized with pIgA treatment if applicable. siRNA sequences are shown in Table 2.

## Cell viability detection

SV40-MES13 cells were uniformly inoculated into a 96-well plate with $1 \times 10^3$ cells/well. After each group was treated separately, the cells were rinsed with sterile PBS twice. Next, CCK-8 working solution (10 μL, Beyotime Biotechnology) was added and then cultured at 37 °C without light for 2 h. The 96-well plate was placed on the enzyme-labeled instrument (ThermoFisher) to read the absorbance value of 450 nm wavelength, and the cell viability was calculated and analyzed.

## Western blot assay

The protein expressions of GLDC (1:500), PCNA (1:500), cyclinD1 (1:10000) in the SV40-MES13 cells were detected by Western blot assay. An appropriate amount of RIPA lysate (Beyotime Biotechnology) was added, and the extracted total protein was quantified using the BCA method (ThermoFisher). The proteins were then subjected to SDS-PAGE, transmembrane, blocked by using 5% skim milk for 2 h, and incubated with primary antibodies overnight at 4 °C. Next day, secondary antibody was added and then incubated for 1 h at room temperature. ECL reagent (ThermoFisher) was added on the PVDF membrane to develop the membrane. β-actin (1:500) was used as an internal control.

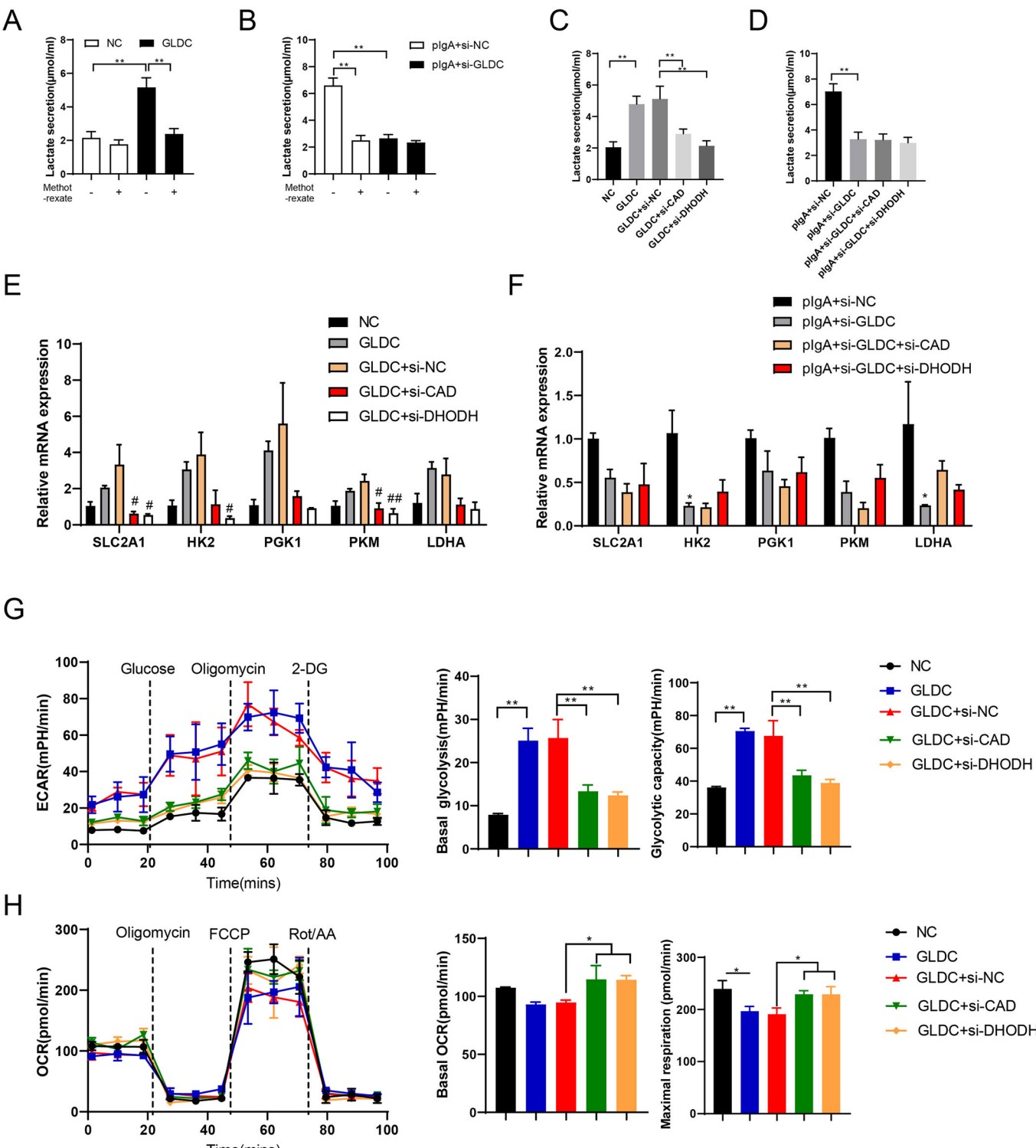

## Cell cycle detection

The variations of cell cycle were detected by flow cytometry. Glomerular mesangial cells in each group were collected, washed twice with PBS, and then resuspended ($5 \times 10^5$ cells/mL). In all, 100 μL cell suspension was taken, the supernatant was discarded by centrifugation at 1000 r/min, and 500 μL binding buffer was added for cell resuspension. Next, propyl iodide (5 μL, Thermo-Fisher) reagents were added, mixed and incubated at room temperature for 15 min away from light, and then the cell cycle was detected by flow cytometry (Beckman Coulter Life Sciences).

**Figure 5. GLDC-regulated pyrimidine synthesis fuels glycolysis in glomerular mesangial cells.**

(A–D) Lactate secretion was detected in the NC, GLDC, NC + Methotrexate, GLDC + Methotrexate, GLDC + si-CAD, GLDC + si-DHODH, plgA + si-NC, plgA + si-GLDC, plgA + si-NC + Methotrexate, plgA + si-GLDC + Methotrexate, plgA + si-GLDC + si-CAD, plgA + si-GLDC + si-DHODH group SV40-MES13 cells. (A) One-way ANOVA with Tukey's post-hoc test, $N = 3$, Lane 1 *vs.* Lane 3, **$P < 0.0001$; Lane 3 *vs.* Lane 4, **$P = 0.0001$. Error bar: mean ± SD. (B) One-way ANOVA with Tukey's post-hoc test, $N = 3$; Lane 1 *vs.* Lane 3, **$P < 0.0001$; Lane 1 *vs.* Lane 3, **$P < 0.0001$. Error bar: mean ± SD. (C) One-way ANOVA with Tukey's post-hoc test, $N = 3$, NC *vs.* GLDC, **$P = 0.0059$; GLDC+si-NC *vs.* GLDC+si-CAD, **$P = 0.0018$; GLDC+si-NC *vs.* GLDC+si-DHODH, **$P = 0.0022$; Error bar: mean ± SD. (D) One-way ANOVA with Tukey's post-hoc test, $N = 3$, plgA+si-NC *vs.* plgA+si-GLDC, **$P = 0.0001$. Error bar: mean ± SD. (E, F) The mRNA levels of SLC2A1, HK2, PGK1, PKM, and LDHA were detected by RT-qPCR in the NC, GLDC, GLDC + si-NC, GLDC + si-CAD, GLDC + si-DHODH, plgA + si-NC, plgA + si-GLDC, plgA + si-GLDC + si-CAD, plgA + si-GLDC + si-DHODH group SV40-MES13 cells. (E) One-way ANOVA with Tukey's post-hoc test, $N = 3$, SLC2A1 mRNA: GLDC+si-NC *vs.* GLDC+si-CAD, #$P = 0.0233$; GLDC+si-NC *vs.* GLDC+si-DHODH, #$P = 0.0198$. HK2 mRNA: GLDC+si-NC *vs.* GLDC+si-DHODH, #$P = 0.0294$. PKM mRNA: GLDC+si-NC *vs.* GLDC+si-CAD, #$P = 0.0165$; GLDC+si-NC *vs.* GLDC+si-DHODH, **$P = 0.0057$. Error bar: mean ± SEM. (F) One-way ANOVA with Tukey's post-hoc test, $N = 3$, HK2 mRNA: plgA+si-NC *vs.* plgA+si-GLDC, *$P = 0.018$. LDHA mRNA: plgA+si-NC *vs.* plgA+si-GLDC, *$P = 0.0223$. Error bar: mean ± SEM. (G, H) ECAR and OCR were detected in the NC, GLDC, GLDC + si-NC, GLDC + si-CAD, GLDC + si-DHODH group HMCs. (G, left) One-way ANOVA with Tukey's post-hoc test, $N = 3$, NC *vs.* GLDC, **$P < 0.0001$; GLDC+si-NC *vs.* GLDC+si-CAD, **$P = 0.0007$; GLDC+si-NC *vs.* GLDC+si-DHODH, **$P = 0.0004$. (G, right) One-way ANOVA with Tukey's post-hoc test, $N = 3$, NC *vs.* GLDC, **$P < 0.0001$; GLDC+si-NC *vs.* GLDC+si-CAD, **$P = 0.0005$; GLDC+si-NC *vs.* GLDC+si-DHODH, **$P = 0.0001$. (H, left) One-way ANOVA with Tukey's post-hoc test, $N = 3$, GLDC+si-NC *vs.* GLDC+si-CAD, *$P = 0.0112$; GLDC+si-NC *vs.* GLDC+si-DHODH, *$P = 0.0123$. (H, right) One-way ANOVA with Tukey's post-hoc test, $N = 3$, NC *vs.* GLDC, *$P = 0.0102$; GLDC+si-NC *vs.* GLDC+si-CAD, *$P = 0.0206$; GLDC+si-NC *vs.* GLDC+si-DHODH, *$P = 0.0216$. Error bar: mean ± SD. Source data are available online for this figure.

## Enzyme-linked immunosorbent assay (ELISA)

The levels of MCP-1, interleukin-6 (IL-6), complement 3 (C3), TGF-β1, and TNF-α in cell supernatant were detected ELISA assay. The cell culture medium of each group was collected, and the supernatant was obtained by centrifugation. The contents of cell supernatant MCP-1, IL-6, C3, TGF-β1, and TNF-α were detected according to the instructions of MCP-1 ELISA kit (Beyotime Biotechnology), IL-6 ELISA kit (Beyotime Biotechnology), C3 ELISA kit (Elabscience), TGF-β1 ELISA kit (Elabscience), and TNF-α ELISA kit (Elabscience). The absorbance of each well was measured at 450 nm by an enzyme-labeled instrument (ThermoFisher).

## Liquid chromatography-mass spectrometry (LC/MS) non-target metabolomics analysis

The glomerulus sample and 500 μL 70% methanol/water mixture were added into a 2 mL centrifuge tube, and homogenized by vortexing for 3 min and ultrasonic for 10 min. After centrifugation at 12,000 r/min for 10 min, 300 μL of supernatant was transferred to a new centrifuge tube, placed in a refrigerator at −20 °C for 30 min, and then centrifuged at 4 °C and 12,000 r/min for 10 min. LC-MS analysis was performed by transferring 200 μL supernatant through a protein precipitation plate. Cells in each group with good growth state were selected for cell count, and the number of cell samples in each group was controlled at about $1 \times 10^7$. The cells in the dish were then transferred to a centrifuge tube for further LC-MS analysis.

Vanquish UHPLC System instrument (ThermoFisher Scientific) was used for the analysis on ACQUITY UPLC® HSS T3 column (1.8 μm, 2.1 × 100 mm) (Waters). The temperature of automatic sampler was 8 °C, the flow rate was 0.3 mL/min, the column temperature was 40 °C, and 2 μL samples were injected for gradient elution. For LC-ESI ( + )-MS analysis, the procedure was set as follows: 0–1 min, 8% B2 [0.1% formic acid in acetonitrile (v/v)]; 1–8 min, 8%–98% B2; 8–10 min, 98% B2; 10–10.1 min, 98%–8% B2; 10.1–12 min, 8% B2. For LC-ESI (-)-MS analysis, the gradient elution procedure is the same, but the mobile phase is acetonitrile (B3) and ammonium formate (5 mM, A3). Q Exactive

instrument (ThermoFisher Scientific), electron spray ionization (ESI), positive and negative ion ionization mode were used for mass spectrometry. The positive ion spray voltage (3.50 kV), the negative ion spray voltage (2.50 kV), the sheath gas pressure (40 arb), and the aux gas flow (10 arb) were set. Capillary temperature was 325 °C, MS1 resolving power was 70,000 FWHM, range 100–1000 $m/z$, normalized collision energy was 30 eV, and dynamic exclusion eliminated unnecessary MS/MS information.

## Measurement of metabolites secretion and lactate generation

The contents of glycine were detected by ELISA assay according to the manufacturer's instructions of corresponding ELISA Kit (CLOUD-CLONE CORP. WUHAN). The absorbance of each sample was measured at 450 nm by using an enzyme-labeled instrument (Beckman Coulter Life Sciences).

The production of aerobic metabolites in the harvested culture medium of SV40-MES13 cells was evaluated by lactate release assay using lactic acid detection kit (Beijing Solarbio Science & Technology Co., Ltd.). The absorbance of the sample at 570 nm wavelength was measured by enzyme-labeled instrument.

## Measurement of extracellular acidification ratio (ECAR), oxygen consumption ratio (OCR)

HMCs were seeded at $1 \times 10^4$ cells/mL in Seahorse Xfe-24 culture plate and cultured for 24 h. According to the Seahorse XF Glycolytic Rate Assay Kit (Agilent Technologies) instructions, glucose (10 mmol/L), oligomycin (1 μmol/L) and 2-DG (100 mmol/L) were added and culture for 30 min, and the ECAR was detected by Extracellular Flux Analyzer (XFe24, Agilent Technologies). According to the instructions of the Seahorse XFp Cell Mito Stress Test Kit (Agilent Technologies), oligomycin (1.5 μmol/L), mito-chondrial uncoupler (1 μmol/L), rotenone/antimycin A (0.5 μmol/L) were added. The OCR of the cells was examined after 0.5 h incubation by Extracellular Flux Analyzer (XFe24, Agilent Technologies).

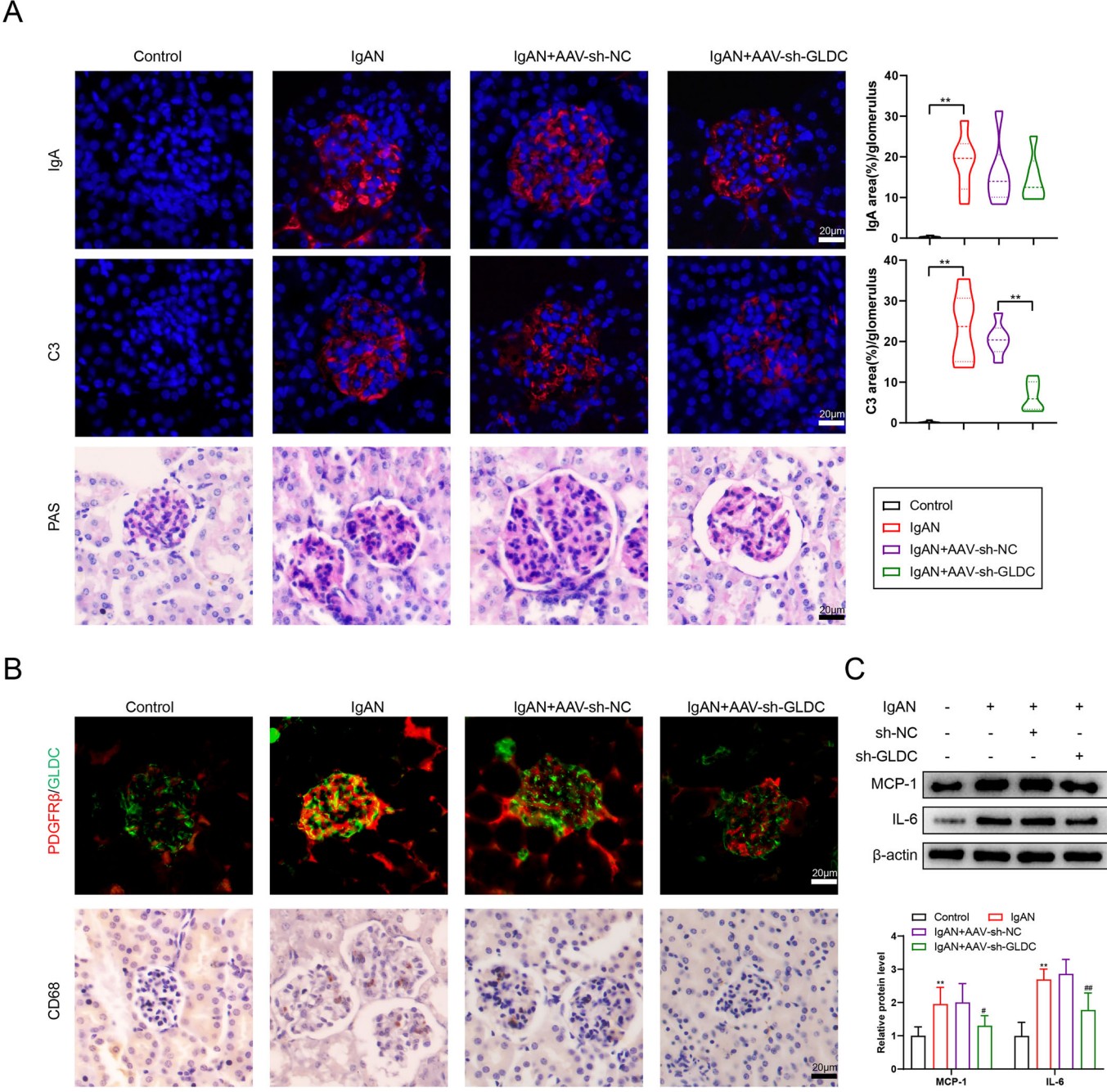

**Figure 6. Silencing GLDC in vivo alleviated IgAN progression.**

Mice were grouped into control, IgAN, IgAN + AAV-sh-NC, IgAN + AAV-sh-GLDC (N = 6/group). (**A**) The positive expressions of IgA and C3 in the glomeruli were detected by immunohistochemical staining assay. PAS staining was performed. Scale bar = 20 μm. (Upper) One-way ANOVA with Tukey's post-hoc test, N = 6, Control *vs.* IgAN, **P = 0.0003. (Lower) Brown–Forsythe ANOVA with Games–Howell's post-hoc test, N = 6, Control *vs.* IgAN, **P = 0.0042; IgAN+AAV-sh-NC *vs.* IgAN+AAV-sh-GLDC, **P = 0.0005. (**B**) Immunohistochemical staining of GLDC/PDGFRβ and CD68 were performed. Scale bar = 20 μm. (**C**) Western blot assay was used to detect MCP-1 and IL-6 protein expressions. One-way ANOVA with Tukey's post-hoc test, N = 6, MCP-1: Control *vs.* IgAN, **P = 0.0048; IgAN+AAV-sh-NC *vs.* IgAN+AAV-sh-GLDC, #P = 0.0467. IL-6: Control *vs.* IgAN, **P < 0.0001; IgAN+AAV-sh-NC *vs.* IgAN+AAV-sh-GLDC, ##P = 0.0013. Error bar: mean ± SD. Source data are available online for this figure.

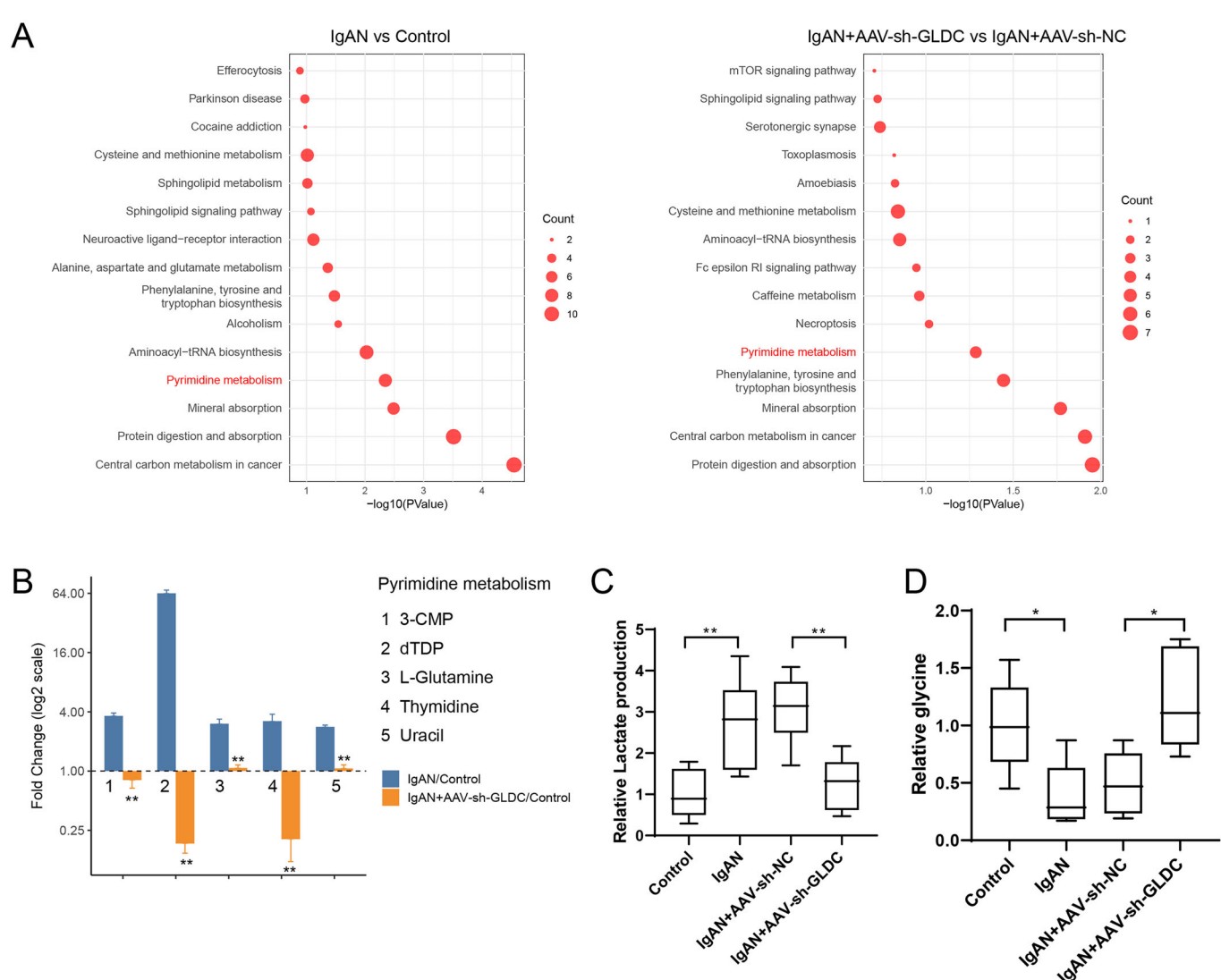

**Figure 7. Silencing GLDC in vivo suppressed pyrimidine metabolism and glycolysis in glomeruli.**

Mice were grouped into control, IgA, IgAN + AAV-sh-NC, IgAN + AAV-sh-GLDC (N = 6/group). (A) LC/MS non-target metabolomics analysis and KEGG pathway enrichment analysis were performed (control vs IgAN, IgAN + AAV-sh-NC vs IgAN + AAV-sh-GLDC). (B) 3-CMP, dTDP, ʟ-glutamine, thymidine, and uracil levels were detected by ELISA. Unpaired *t* test, N = 6, No. 1, **P < 0.0001; No. 2, **P < 0.0001; No. 3, **P = 0.0003; No. 4, **P = 0.0002; No. 5, **P < 0.0001. Error bar: mean ± SEM. (C, D) Lactate secretion and glycine was detected. (C) One-way ANOVA with Tukey's post-hoc test, N = 6, Control *vs.* IgAN, **P = 0.008; IgAN+AAV-sh-NC *vs.* IgAN +AAV-sh-GLDC, **P = 0.0053. Three horizontal lines within the box (from top to bottom) represent the 75th percentile, 50th percentile, and 25th percentile of the data, respectively. (D) One-way ANOVA with Tukey's post-hoc test, N = 6, Control *vs.* IgAN, *P = 0.0325; IgAN+AAV-sh-NC *vs.* IgAN+AAV-sh-GLDC, *P = 0.0107. Three horizontal lines within the box (from top to bottom) represent the 75th percentile, 50th percentile, and 25th percentile of the data, respectively. Source data are available online for this figure.

## Real time quantitative PCR (RT-qPCR)

Total RNA was extracted using the RNA isolator total RNA extraction reagent (Vazyme Biotech Co., Ltd.), and mRNA was reverse-transcribed using the HiScript III RT SuperMix reagent (Vazyme Biotech Co., Ltd.). Then, ChamQ Universal SYBR qPCR Master Mix agent was used to detect the mRNA levels of SLC2A1, HK2, PGK1, PKM, and LDHA in 7300 Real-Time PCR instrument (Applied Biosystems). Using β-actin as the internal reference, and $2^{-\triangle\triangle CT}$ method calculated the relative expression levels. The primer sequences are shown in Table 3.

## Statistical analysis

The data were statistically analyzed by GraphPad Prism 8.0 and SPSS 25.0 software. Bioinformatics analysis was performed using R package DESeq2, average linkage hierarchical clustering and Spearman's Rank-Order Correlation. Fisher's exact test was used to detect the correlation between GLDC expression level in kidney tissue and clinical data of patients. In addition, two-tailed paired *t* test, one-way ANOVA with Tukey's post-hoc, two-way ANOVA with Sidak's post-hoc, Kruskal–Wallis with Dunn's post-hoc were used. *P* < 0.05 was considered to be statistically

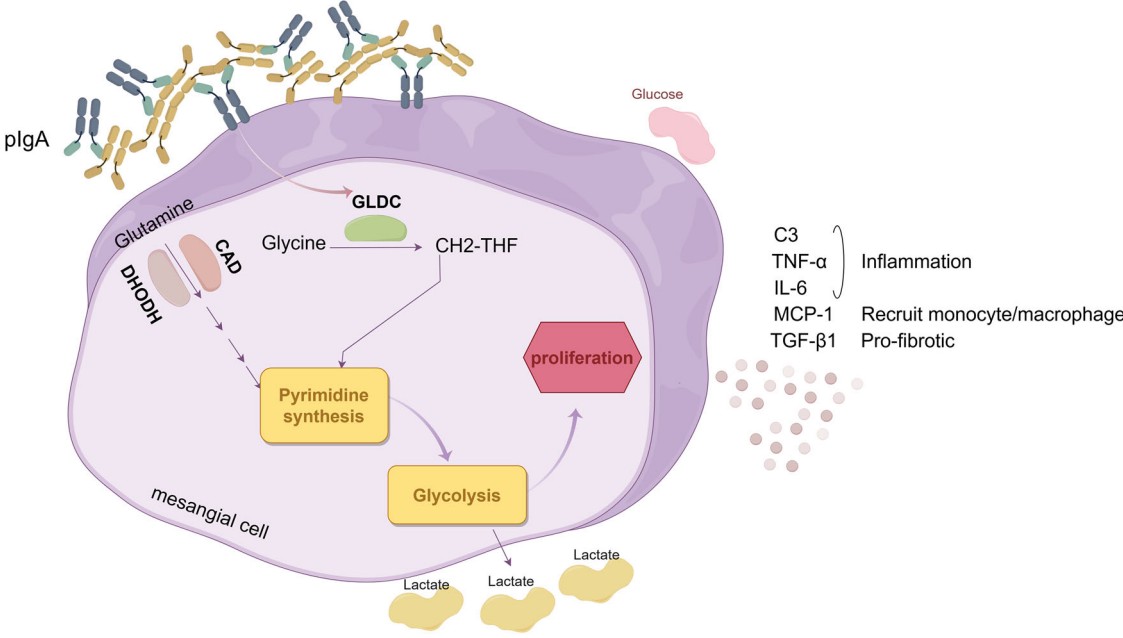

**Figure 8. Under the stimulation of pIgA, the expression of GLDC in glomerular mesangial cells is increased.**

This upregulation of GLDC modulates the glycolytic pathway through pyrimidine metabolism, thereby promoting the proliferation of glomerular mesangial cells. Concurrently, the secretions from glomerular mesangial cells, such as C3, TGF-β1, and MCP-1, are increased. These secreted factors contribute to enhanced inflammatory responses, recruitment of macrophages, and fibrotic reactions in the surrounding interstitium.

**Table 2. siRNA sequences used in this study.**

| siRNAs | Sequences |
| --- | --- |
| si-NC (negative control) | AAGCTTCATAAGGCGCATAGC |
| si-GLDC (mouse) | GTCCAGACTCGAGCCAAATAT |
| si-CAD (mouse) | GTGGCCTCAAGTATAATCAGA |
| si-DHODH (mouse) | GTGGAGGACTTCTCTTCACCT |
| si-GLDC (human) | GTCCAGACTCGAGCCAAATAT |
| si-CAD (human) | GCCAAGTGCTAGTAGACAAGT |
| si-DHODH (human) | GCCAGGATAAGGAGGACATTG |

**Table 3. The primer sequences used in this study.**

| Gene name | Forward/reverse | Sequences |
| --- | --- | --- |
| SLC2A1 | Forward (5'–3') | CACTGTGGTGTCGCTGTTTG |
| | Reverse (5'–3') | ATGGAATAGGACCAGGGCCT |
| HK2 | Forward (5'–3') | CACCCTACAGCAGCTGTGAA |
| | Reverse (5'–3') | TCTCCATCTCCACCCTCTGG |
| PGK1 | Forward (5'–3') | TGGCCTCTGGTATACCTGCT |
| | Reverse (5'–3') | ATGCAACCCCTAGAAGTGGC |
| PKM | Forward (5'–3') | TCACCCTGGACAACGCTTAC |
| | Reverse (5'–3') | CACCATGTCCACATCCTGCT |
| LDHA | Forward (5'–3') | CAAGGAGCAGTGGAAGGAGG |
| | Reverse (5'–3') | CCAAGTCTGCCACAGAGAGG |

**The paper explained**

**Problem**

IgA nephropathy (IgAN) is a common kidney disease characterized by the buildup of IgA immune complexes in glomeruli, leading to inflammation, scarring, and often progressive kidney damage. It can cause symptoms like blood in the urine and proteinuria, and may eventually lead to kidney failure. Current treatments are limited and not always effective, highlighting the need for a deeper understanding of the disease's mechanisms and new therapeutic targets.

**Results**

This study identified glycine decarboxylase (GLDC) as a key molecule involved in IgAN progression. Researchers found that GLDC is highly expressed in the glomeruli of IgAN patients and mice models. In vitro experiments showed that GLDC promotes the proliferation of glomerular mesangial cells, which is a hallmark of IgAN, through the pyrimidine metabolic pathway. This metabolic pathway was found to fuel glycolysis, further enhancing cell proliferation. In vivo experiments in mice confirmed that silencing GLDC expression alleviated IgAN progression by reducing glomerular damage and inflammation.

**Impact**

These findings provide new insights into the pathogenesis of IgAN, highlighting the role of GLDC and metabolic pathways in disease progression. Targeting GLDC or its downstream metabolic pathways could offer novel therapeutic strategies to slow or halt the progression of IgAN, potentially reducing the need for dialysis or kidney transplantation in affected patients.

significant. The experiment in this study was repeated at least three times.

## Data availability

Metabolomics profile of MES-13 exposed to pIgA (S-BSST2149) and glomerulus samples from IgAN mice (S-BSST2150) can be found in websites https://www.ebi.ac.uk/biostudies/studies/S-BSST2149?key=726b97c6-5b7c-42d4-a2b4-fad5ca41b821 and https://www.ebi.ac.uk/biostudies/studies/S-BSST2150?key=8645597b-3ac5-4638-963e-1efa234b0dfa.

The source data of this paper are collected in the following database record: biostudies:S-SCDT-10_1038-S44321-025-00315-2.

## Peer review information

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

## Acknowledgements

This work was supported by the National Natural Science Foundation of China (No. 82260143), the Kidney Disease Engineering Technology Research Centre Foundation of Jiangxi Province (No. 20164BCD40095), the Key Project of Jiangxi Provincial Nature Science Foundation (No. 20224ACB206008), the "Thousand Talents Plan" project of introducing and training high-level talents of innovation and entrepreneurship in Jiangxi Province (No. JXSQ2023201030), the Jiangxi Province Key Laboratory of Molecular Medicine (No. 2024SSY06231), and the Science and Technology Plan of Health Commission of Jiangxi Province (No. 202210619), and the Jiangxi Provincial Natural Science Foundation (No. 20252BAC200465).

## Author contributions

**Yi Xiong**: Conceptualization; Writing—original draft. **Fang Zeng**: Conceptualization; Writing—original draft. **Kaiping Luo**: Formal analysis. **Li Wang**: Formal analysis. **Manna Li**: Formal analysis. **Yanxia Chen**: Data curation. **Tianlun Huang**: Data curation. **Chengyun Xu**: Data curation. **Gaosi Xu**: Conceptualization; Writing—original draft. **Honghong Zou**: Conceptualization; Writing—original draft.

Source data underlying figure panels in this paper may have individual authorship assigned. Where available, figure panel/source data authorship is listed in the following database record: biostudies:S-SCDT-10_1038-S44321-025-00315-2.

## Disclosure and competing interests statement

The authors declare no competing interests.

# Expanded View Figures

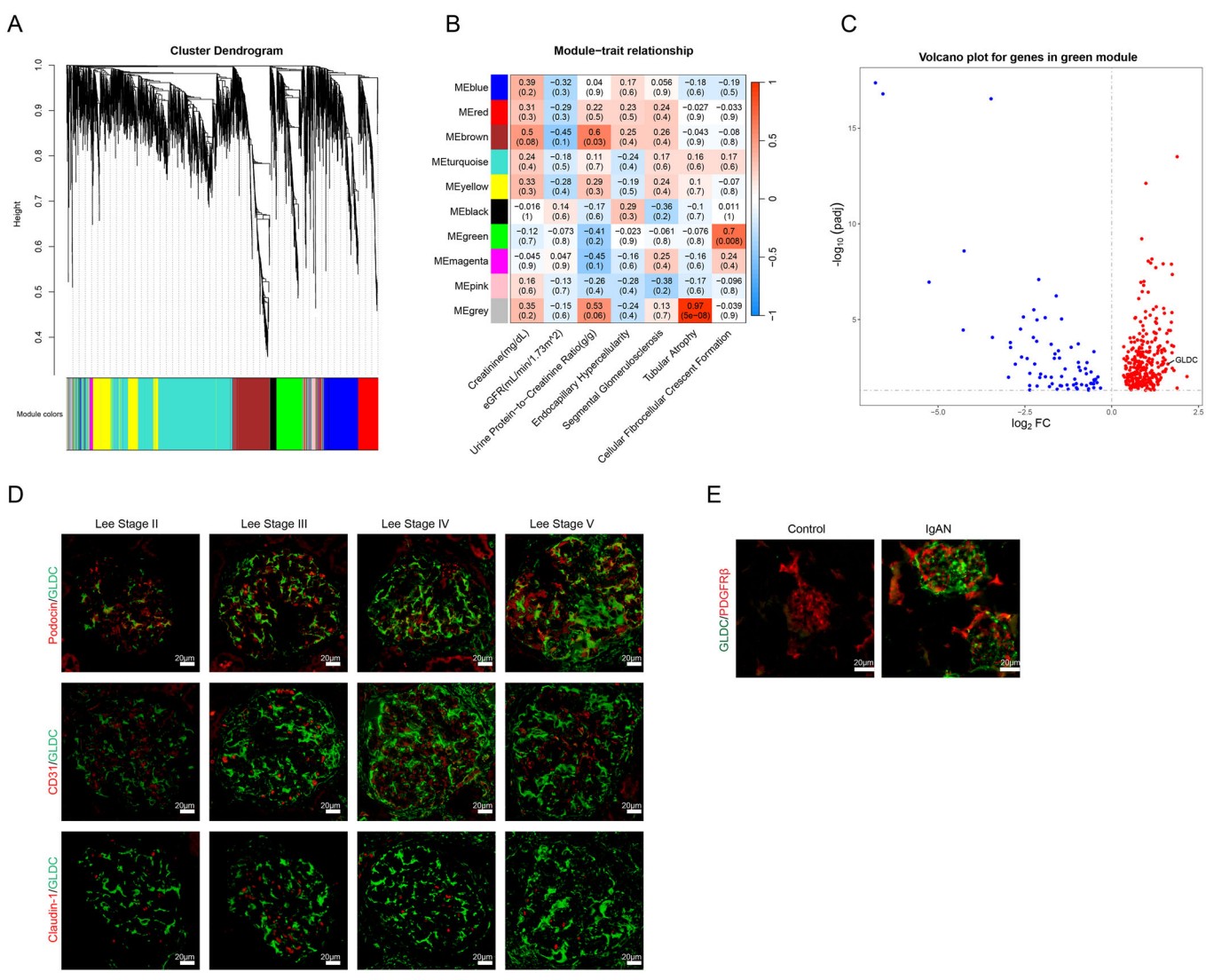

**Figure EV1. Analysis of the correlation between GLDC-containing modules and crescent formation in the clinical IgAN dataset.**

(**A**) The GSE141295 dataset was downloaded for differential expression analysis (IgAN vs Normal). (**B**) The correlation analysis between gene modules and clinical features. (**C**) KEGG pathway enrichment analysis. (**D**) Co-staining of GLDC with CD31 (an endothelial cell marker), podocin (a podocyte marker), and claudin-1 (a parietal epithelial cell marker and a component of crescents). (**E**) The expression of GLDC and PDGFRβ in normal and IgAN mice. Source data are available online for this figure.

A

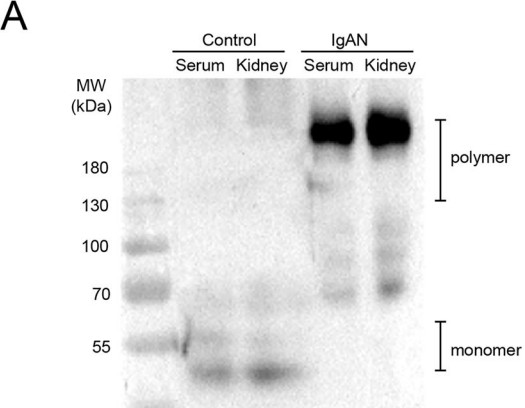

B

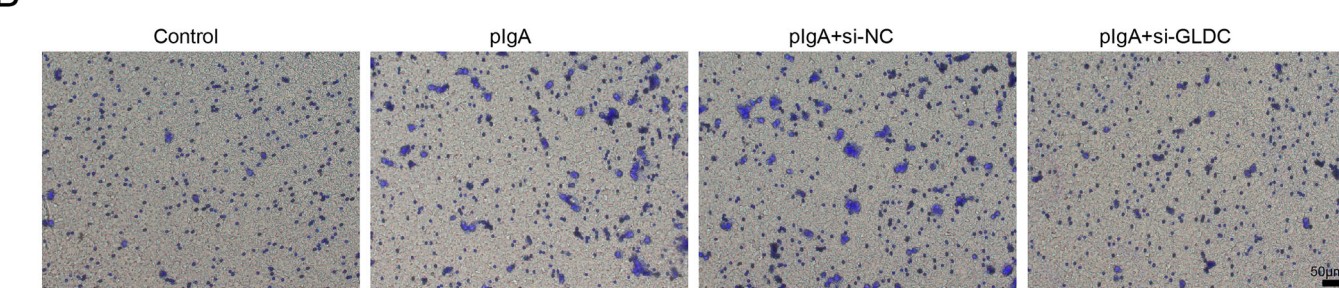

C

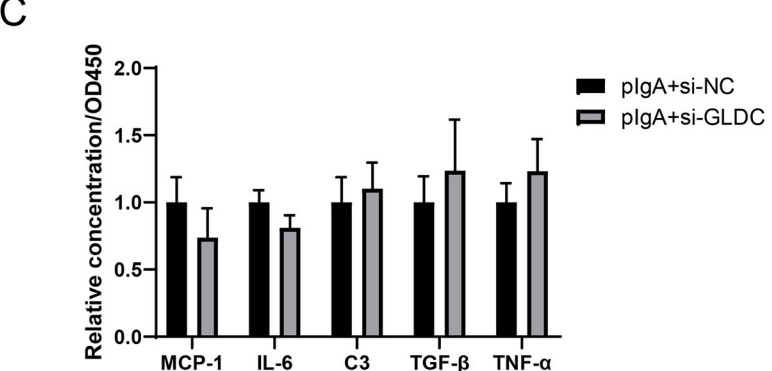

Figure EV2. The effect of GLDC on the chemotactic ability of glomerular mesangial cells.

(A) Expression of IgA in serum and kidney tissue of normal and IgAN model mice. (B) SV40-MES13 cells in the control, pIgA, pIgA + si-NC, pIgA + si-GLDC groups were co-cultured with Raw264.7 derived macrophage, and the chemotactic effects on macrophages were analyzed by Transwell assay. (C) ELISA was used to detect MCP-1, IL-6, C3, TGF-β1, and TNF-α content in the pIgA + si-NC and pIgA + si-GLDC groups. Source data are available online for this figure.

A

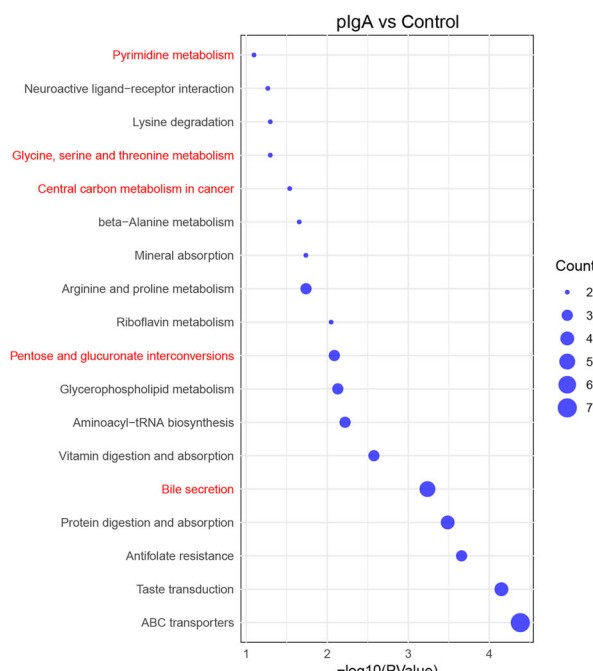

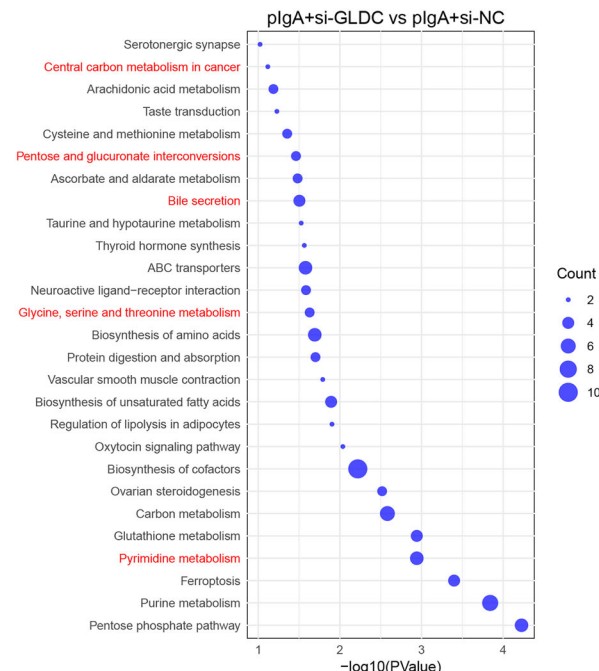

B

C

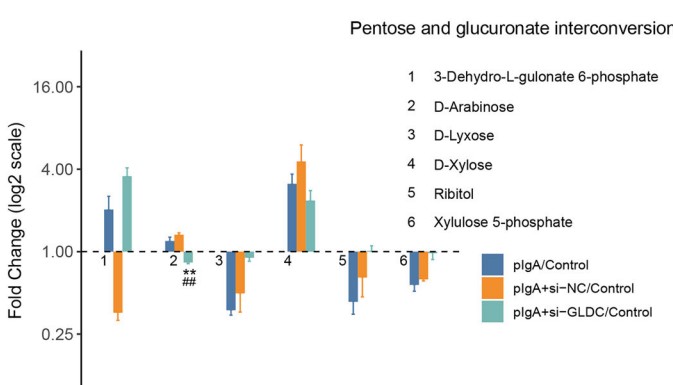

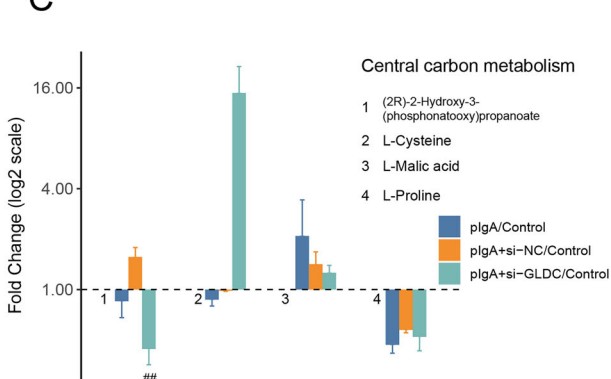

D

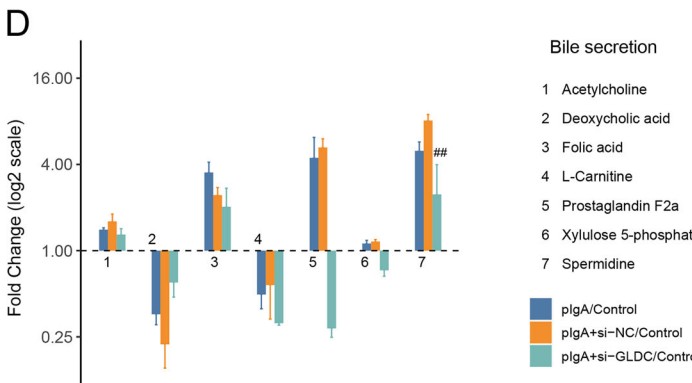

◄ **Figure EV3.  The key metabolic pathways that GLDC regulates glomerular mesangial cell proliferation.**

(**A**) LC/MS non-target metabolomics analysis and KEGG pathway enrichment analysis were performed (control vs pIgA, pIgA + si-NC vs pIgA + si-GLDC). (**B**–**D**) The metabolites of pentose and glucuronate interconversions, central metabolism, bile secretion in the pIgA + si-NC and pIgA + si-GLDC groups were analyzed. *$P < 0.05$. Source data are available online for this figure.

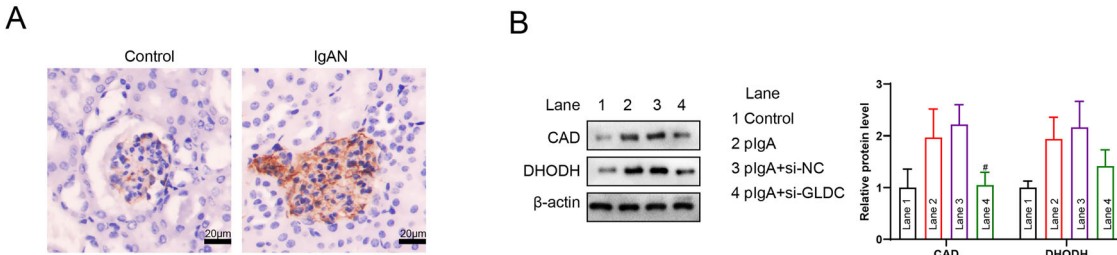

**Figure EV4. The expressions of CAD and DHODH proteins.**

(A) The positive expression of CAD in the glomeruli of control mice ($N = 6$) and IgAN model mice ($N = 6$) were detected by immunohistochemical staining assay.
(B) Western blot assay was used to detect CAD and DHODH protein expressions in the control, pIgA, pIgA + si-NC, pIgA + si-GLDC group SV40-MES13 cells. [#]$P < 0.05$ vs pIgA + si-NC. Source data are available online for this figure.

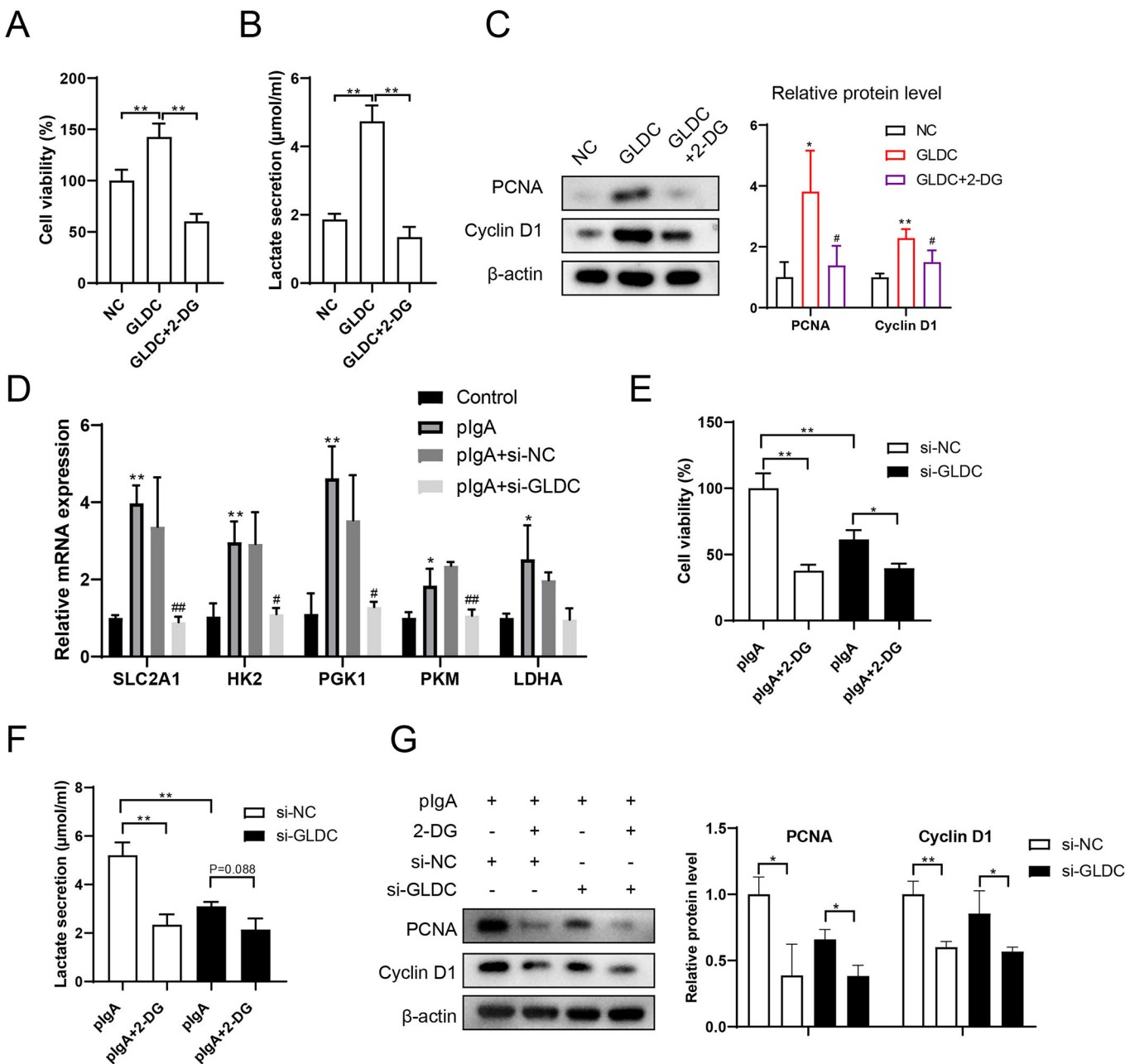

**Figure EV5.** The role of the glycolytic pathway in the regulation of mesangial cell growth by GLDC.

(A–C) SV40-MES13 cells were grouped into NC, GLDC, GLDC + 2-DG. (A) Cell viability assay was performed. **$P < 0.01$. (B) Lactate secretion was detected. **$P < 0.01$. (C) Western blot assay was used to detect Cyclin D1 and PCNA protein expressions. **$P < 0.01$; ##$P < 0.01$. (D) The mRNA levels of SLC2A1, HK2, PGK1, PKM, and LDHA were detected by qPCR in the control, pIgA, pIgA + si-NC, pIgA + si-GLDC group "-->SV40-MES13 cells. *$P < 0.05$, **$P < 0.01$; #$P < 0.05$, ##$P < 0.01$. (E–G) SV40-MES13 cells were grouped into pIgA + si-NC, pIgA + si-GLDC, pIgA + si-NC + 2-DG, pIgA + si-GLDC + 2-DG. (E) Cell viability assay was performed. *$P < 0.05$, **$P < 0.01$. (F) Lactate secretion was detected. **$P < 0.01$. (G) Western blot assay was used to detect Cyclin D1 and PCNA protein expressions. **$P < 0.01$. Source data are available online for this figure.

