## [Peer Review File · EMBO Molecular Medicine]

Glycine decarboxylase advances IgA nephropathy by boosting mesangial cell proliferation through the pyrimidine pathway

Yi Xiong, Fang Zeng, Kaiping Luo, Li Wang, Manna Li, Yanxia Chen, Tianlun Huang, Chengyun Xu, Gaosi Xu, and Honghong Zou

Corresponding authors: Honghong Zou (ndefy21195@ncu.edu.cn) , Gaosi Xu (ndefy08027@ncu.edu.cn)

Review Timeline:

Submission Date:	31st Mar 25
Editorial Decision:	21st May 25
Revision Received:	19th Aug 25
Editorial Decision:	2nd Sep 25
Revision Received:	14th Sep 25
Accepted:	16th Sep 25

Editor: Zeljko Durdevic

Transaction Report:

21st May 2025

Dear Dr. Zou,

Thank you for the submission of your manuscript to EMBO Molecular Medicine, and please accept my apologies for the delay in getting back to you, which is due to the fact that one referee needed more time to complete his/her review. We have now received feedback from the two reviewers who agreed to evaluate your manuscript.

As you will see from the reports pasted below, both referees recognize interest of the study but also raise important concerns that should be addressed in a major revision. After the cross-commenting discussion it became clear that referee #1 point 1 should be addressed by discussion and point 2 through stratified analyses or statistical adjustment for sex as a covariate. If you would like to discuss further the points raised by the referees, I am available to do so via email or video. Let me know if you are interested in this option.

Further consideration of a revision that addresses reviewers' concerns in full will entail a second round of review. EMBO Molecular Medicine encourages a single round of revision only and therefore, acceptance or rejection of the manuscript will depend on the completeness of your responses included in the next, final version of the manuscript. For this reason, and to save you from any frustrations in the end, I would strongly advise against returning an incomplete revision. Further, when submitting the revised manuscript please be sure to add institutional email addresses for Honghong Zou in the manuscript and our submission system

We would welcome the submission of a revised version within three months for further consideration. Please let us know if you require longer to complete the revision.

I look forward to receiving your revised manuscript.

Yours sincerely,

Zeljko Durdevic

Zeljko Durdevic
Senior Editor
EMBO Molecular Medicine

We require:

- 1) A .docx formatted version of the manuscript text (including legends for main figures, EV figures and tables). Please make sure that the changes are highlighted to be clearly visible.
- 2) Individual production quality figure files as .eps, .tif, .jpg (one file per figure). For guidance, download the 'Figure Guide PDF': (<https://www.embopress.org/page/journal/17574684/authorguide#figureformat>).
- 3) A .docx formatted letter INCLUDING the reviewers' reports and your detailed point-by-point responses to their comments. As part of the EMBO Press transparent editorial process, the point-by-point response is part of the Review Process File (RPF), which will be published alongside your paper.
- 4) A complete author checklist, which you can download from our author guidelines (<https://www.embopress.org/page/journal/17574684/authorguide#submissionofrevisions>). Please insert information in the

checklist that is also reflected in the manuscript. The completed author checklist will also be part of the RPF.

6) It is mandatory to include a 'Data Availability' section after the Materials and Methods. Before submitting your revision, primary datasets produced in this study need to be deposited in an appropriate public database, and the accession numbers and database listed under 'Data Availability'. Please remember to provide a reviewer password if the datasets are not yet public (see <https://www.embopress.org/page/journal/17574684/authorguide#dataavailability>).

12) Author contributions: You will be asked to provide CRediT (Contributor Role Taxonomy) terms in the submission system. These replace a narrative author contribution section in the manuscript.

13) A Conflict of Interest statement should be provided in the main text.

14) Every published paper now includes a 'Synopsis' to further enhance discoverability. Synopses are displayed on the journal webpage and are freely accessible to all readers. They include a short stand first (maximum of 300 characters, including space) as well as 2-5 one-sentences bullet points that summarizes the paper. Please write the bullet points to summarize the key NEW findings. They should be designed to be complementary to the abstract - i.e. not repeat the same text. We encourage inclusion of key acronyms and quantitative information (maximum of 30 words / bullet point). Please use the passive voice. Please attach these in a separate file or send them by email, we will incorporate them accordingly.

15) Include a Reagents and Tools Table as part of the Methods section, which can be downloaded from our author guidelines (<https://www.embopress.org/page/journal/17574684/authorguide#structuredmethods>)

***** Reviewer's comments *****

Referee #1 (Remarks for Author):

In this study, the authors aimed to investigate whether glycine decarboxylase (GLDC) contributes to mesangial cell proliferation in IgA nephropathy (IgAN). The study was generally well designed and conducted; however, the following suggestions are offered for improvement:

1. GLDC is involved in pyrimidine metabolism and would be involved in proliferation in any cell. In other words, it would be involved in any disease that causes cell proliferation; it may not be specifically involved in the pathogenesis of IgAN.
2. Regarding the clinical information, there is a significant difference in the number of primary IgAN patients between males and females as well as across the different Lee stages.
3. Regarding the bioinformatics analysis, the authors used bulk RNA-seq, and the corresponding Lee stage information is necessary to enable comparison with the results shown in Figure 1A. Moreover, it remains unclear which specific cell types are affected by the GLDC-related pathways mentioned, given the limitations of bulk RNA-seq. In addition, both PDGFR β - and GLDC-positive areas in Figure 1A appear to include extensive non-mesangial regions, whereas only a limited number of mesangial regions seem to be double-positive. If this interpretation is incorrect, appropriate negative controls should be provided to clarify the findings. Additionally, regarding Figure 1C, the IF images of GLDC should be accompanied by mesangial cell marker staining to more clearly demonstrate its localization and clarify whether GLDC is specifically expressed in mesangial cells.
4. There are some typographical errors in lines 78, 79, and 80 that should be corrected.
- 5 . Regarding Ref. 36, isn't it a beta II spectrin, not a beta III spectrin?

Referee #2 (Remarks for Author):

The authors validated through in vivo and in vitro experiments that GLDC promotes mesangial cell proliferation via the pyrimidine pathway, thereby accelerating the progression of IgA nephropathy. The study demonstrates a well-founded experimental rationale, comprehensive design, and clear result figures with notable scientific innovation. However, several limitations should be addressed: 1) Inaccuracies were identified in the description of certain methodological details; 2) The result figures lack sufficient explanatory annotations to ensure self-explanatory clarity; 3) The Western Blot analysis requires submission of original raw data images from at least three independent experimental replicates. These revisions are essential to strengthen methodological transparency and data reproducibility, therefore, the manuscript is recommended for acceptance pending adequate revision of the aforementioned issues.

Suggestions :

1. The abstract does not adequately introduce the research objective, methodology, key findings, or significance of the study, making it difficult to intuitively grasp the article's content. It is recommended to revise and expand the abstract to address these elements.
2. In the Methods section (2.7 Cell Culture and Treatment), the primer sequences should be compiled into a table for clarity and

ease of reference.

3. All figure panels lack descriptive legends and annotations at the bottom, compromising the self-explanatory clarity of the results. Detailed captions and labels should be supplemented to enhance interpretability.

4. A quantitative analysis of all Western Blot images is strongly advised. Additionally, raw data images from at least three independent experimental replicates must be provided to ensure reproducibility and statistical validity.

5. In the Discussion section, incorporating a mechanistic diagram to summarize the experimental validation of the study would enhance the article's visual clarity and scientific completeness.

Dear Editor:

Thank you very much for your suggestion and advice. We also would like to thank the reviewers for their valuable comments. We have thoroughly considered the comments and substantially revised the manuscript. We hope this version is much improved. A point-to-point response to the comments is given below.

Referee #1 (Remarks for Author):

1. GLDC is involved in pyrimidine metabolism and would be involved in proliferation in any cell. In other words, it would be involved in any disease that causes cell proliferation; it may not be specifically involved in the pathogenesis of IgAN.

Response: The fundamental pathology of IgAN is the proliferation of glomerular mesangial cells^[1]. These proliferating mesangial cells release a series of pro-inflammatory and fibrogenic cytokines that mediate oxidative stress and complement activation, such as C3, CFH, FN1, and TGF-beta1^[2-4]. This process is accompanied by matrix expansion, which leads to podocyte and proximal tubular epithelial cell injury, thereby promoting the progression of IgAN^[5]. Additionally, glomerular mesangial cells can recruit macrophage infiltration through the secretion of inflammatory molecules such as MCP-1, thus affecting the formation of glomerular crescents^[6, 7]. We have also found that the proliferation of glomerular mesangial cells leads to increased release of factors such as MCP-1, IL-6, and C3 (Figure S2C). Therefore, the abnormal proliferation of glomerular mesangial cells is a key pathological change occurring in the glomeruli of IgAN and is worthy of further investigation.

Both bulk RNA-seq analysis of glomeruli and immunohistochemical staining of clinical samples reveal a pathological increase in GLDC in glomeruli, particularly in glomerular mesangial cells. Moreover, clinical data analysis has uncovered a

correlation between GLDC and clinical pathology. This suggests that GLDC may be a potential therapeutic target for IgAN.

Although GLDC is generally involved in cell proliferation, it remains unknown whether GLDC participates in the abnormal proliferation of glomerular mesangial cells in IgAN. Furthermore, the mechanisms by which GLDC affects this abnormal proliferation in IgAN are also unclear. Only through experimental evidence that elucidates the regulatory mechanisms of glomerular mesangial cell proliferation can we recognize the potential importance of GLDC as a therapeutic target for IgAN. For the first time, we have discovered that under pathological conditions of IgAN, glomerular mesangial cells undergo metabolic reprogramming, with pyrimidine metabolism being one of the key altered metabolic pathways. GLDC, as a glycine metabolic enzyme, happens to be a key factor in this regulatory pathway. Under pIgA stimulation, glomerular mesangial cells enhance pyrimidine metabolism by upregulating GLDC expression, thereby affecting the proliferative capacity of glomerular mesangial cells.

As for other regulatory pathways of GLDC and the reasons for its increase under pathological conditions, we are continuing to explore these aspects. To translate our findings into clinical applications, it is essential to fully elucidate the molecular mechanisms of GLDC in disease development and its therapeutic targeting potential to ensure its potential efficacy in clinical treatment.

2. Regarding the clinical information, there is a significant difference in the number of primary IgAN patients between males and females as well as across the different Lee stages.

Response: We further collected 30 additional IgAN samples, and the final statistics are presented in revised Figure 1B. Among these samples, there were 24 male and 25 female cases. The Lee grading classification included 13 cases of Type II, 13 cases of Type III, 12 cases of Type IV, and 11 cases of Type V, in order to achieve as balanced

a distribution of sample types as possible. The result showed that the expression of GLDC was significantly correlated with Lee's grade, segmental glomerulosclerosis (S) and tubular atrophy/interstitial fibrosis (T).

3. Regarding the bioinformatics analysis, the authors used bulk RNA-seq, and the corresponding Lee stage information is necessary to enable comparison with the results shown in Figure 1A. Moreover, it remains unclear which specific cell types are affected by the GLDC-related pathways mentioned, given the limitations of bulk RNA-seq. In addition, both PDGFR β - and GLDC-positive areas in Figure 1A appear to include extensive non-mesangial regions, whereas only a limited number of mesangial regions seem to be double-positive. If this interpretation is incorrect, appropriate negative controls should be provided to clarify the findings. Additionally, regarding Figure 1C, the IF images of GLDC should be accompanied by mesangial cell marker staining to more clearly demonstrate its localization and clarify whether GLDC is specifically expressed in mesangial cells.

Response: Previously, the PDGFR β antibody we used lacked specificity in clinical human samples. Therefore, we replaced it with PDGFR β and GLDC antibodies from another reagent supplier for co-staining of human clinical samples and analyzed the results (see revised Figure 1A). To clarify the co-localization of GLDC in glomeruli, we performed co-staining with CD31 (an endothelial cell marker), podocin (a podocyte marker), and claudin-1 (a parietal epithelial cell marker and a component of crescents) (see revised Figure S1C). We found that GLDC partially co-localizes with podocin, but not significantly with CD31 or claudin-1. To further compare the co-localization coefficients of GLDC with podocin and PDGFR β , we used ImageJ for co-localization analysis. The co-localization coefficient with PDGFR β was significantly higher than that with podocin (see revised Figure 1A), indicating that GLDC is primarily localized in glomerular mesangial cells. Additionally, we supplemented the co-localization detection of PDGFR β and GLDC in glomerular mesangial cells of IgAN mice in revised Figure S1E. In normal mice, the expression

of GLDC in glomeruli is very weak, while in the disease mouse model, the expression is increased and co-localization sites with PDGFR β are observed (revised Figure S1E & 6B).

4. There are some typographical errors in lines 78, 79, and 80 that should be corrected.

Response: We have corrected the errors.

5 . Regarding Ref. 36, isn't it a beta II spectrin, not a beta III spectrin?

Response: It should be anti- β II-spectrin. We have corrected the error.

Referee #2 (Remarks for Author):

1. The abstract does not adequately introduce the research objective, methodology, key findings, or significance of the study, making it difficult to intuitively grasp the article's content. It is recommended to revise and expand the abstract to address these elements.

Response: Thank you for the comments. We have revised the Abstract section.

2. In the Methods section (2.7 Cell Culture and Treatment), the primer sequences should be compiled into a table for clarity and ease of reference.

Response: Thank you for the comments. We have compiled the sequences into tables. Please refer to the revised Table 1 and 2.

3. All figure panels lack descriptive legends and annotations at the bottom,

compromising the self-explanatory clarity of the results. Detailed captions and labels should be supplemented to enhance interpretability.

Response: Thank you for the comments. We have revised figure legends and provided more details.

4. A quantitative analysis of all Western Blot images is strongly advised. Additionally, raw data images from at least three independent experimental replicates must be provided to ensure reproducibility and statistical validity.

Response: Thank you for the comments. We have provided the quantitative analysis of all Western Blot images and raw data images.

5. In the Discussion section, incorporating a mechanistic diagram to summarize the experimental validation of the study would enhance the article's visual clarity and scientific completeness.

Response: Thank you for the comments. We have provided a mechanistic diagram in the discussion section.

References

1. Roberts IS: Pathology of IgA nephropathy. *Nat Rev Nephrol* 2014, 10(8):445-454.
2. Schmitt R, Ståhl AL, Olin AI, Kristoffersson AC, Rebetz J, Novak J, Lindahl G, Karpman D: The combined role of galactose-deficient IgA1 and streptococcal IgA-binding M Protein in inducing IL-6 and C3 secretion from human mesangial cells: implications for IgA nephropathy. *J Immunol* 2014, 193(1):317-326.
3. van den Dobbelaert ME, Verhasselt V, Kaashoek JG, Timmerman JJ, Schroeijers WE, Verweij CL, van der Woude FJ, van Es LA, Daha MR: Regulation of C3 and factor H synthesis of human glomerular mesangial cells by IL-1 and interferon-gamma. *Clin Exp Immunol* 1994, 95(1):173-180.
4. Ebefors K, Liu P, Lassén E, Elvin J, Candemark E, Levan K, Haraldsson B,

- Nyström J: Mesangial cells from patients with IgA nephropathy have increased susceptibility to galactose-deficient IgA1. *BMC Nephrol* 2016, 17:40.
5. Zhao L, Lan Z, Peng L, Wan L, Liu D, Tan X, Tang C, Chen G, Liu H: Triptolide promotes autophagy to inhibit mesangial cell proliferation in IgA nephropathy via the CARD9/p38 MAPK pathway. *Cell Prolif* 2022, 55(9):e13278.
 6. Urushihara M, Kondo S, Kinoshita Y, Ozaki N, Jamba A, Nagai T, Fujioka K, Hattori T, Kagami S: (Pro)renin receptor promotes crescent formation via the ERK1/2 and Wnt/ β -catenin pathways in glomerulonephritis. *Am J Physiol Renal Physiol* 2020, 319(4):F571-f578.
 7. Lai PC, Chiu LY, Srivastava P, Trento C, Dazzi F, Petretto E, Cook HT, Behmoaras J: Unique regulatory properties of mesangial cells are genetically determined in the rat. *PLoS One* 2014, 9(10):e111452.

2nd Sep 2025

Dear Dr. Zou,

Thank you for the submission of your revised manuscript to EMBO Molecular Medicine. I am pleased to inform you that we will be able to accept your manuscript pending the following final amendments:

1) Authors: Please provide institutional email address for both corresponding authors.

2) Figures:

- The table in Figure 1B should be presented as an actual table. So, please remove it from the figure and add it to the main manuscript file as a table with a short title. All main tables should be placed in the manuscript file. Please update the callout in the main text.

- All supplementary figures should be renamed to "Figure EV1" etc. and uploaded as individual, high-resolution figure files. The legends should be added to the manuscript text, after the tables, under the heading "Expanded View Figure Legends". Please update their callouts in the main text.

3) Source data:

- Please upload source data checklist.

- Please upload source data files as one zipped file per figure for the main figures and for the EV figures all source data folders should be uploaded as one zipped file.

4) Please address all comments suggested by our data editors listed below:

o Figure legends:

1. Please note that the legend for figure 2 is not provided in a sequential manner. This needs to be rectified.

2. Please note that the exact p values are not provided in the legends of figures 1A, 2A-J; 3B, C; 4A-J; 5A-H; 6A, C; 7B-D.

3. Please indicate the statistical test used for data analysis in the legends of figures 2A-J; 3B, C; 4A-J; 5A-H; 6A, C.

4. Please note that the box plots need to be defined in terms of minima, maxima, centre, bounds of box and whiskers, and percentile in the legends of figures 1A, 7C, D.

5. Please note that information related to n is missing in the legends of figures 4A-H.

6. Please note that the error bars are not defined in the legends of figures 1A, 2A-J; 3B, C; 4A-J; 5A-H; 6C, 7B.

- Please make sure that all figure callouts are correct and updated.

- Correct order of manuscript sections: Abstract / Keywords / The Paper Explained / Introduction / Results / Discussion / Methods / Data Availability / Acknowledgements / Disclosure and Competing Interests Statement / References / Main Figure Legends / Tables / Expanded View Figure Legends.

- In Methods, provide the statement that informed consent was obtained from all human subjects and confirm that the experiments conformed to the principles set out in the WMA Declaration of Helsinki and the Department of Health and Human Services Belmont Report.

- In Methods, it seems that in "Clinical information" the information about the patients added during revision is missing. Please update this paragraph.

- In Methods, provide the antibody dilutions that were used for each antibody.

- Indicate in legends exact n and exact p values, not a range, along with the statistical test used. To keep the figures "clear" some authors found providing an Appendix table Sx with all exact p-values preferable. You are welcome to do this if you want to.

- In Methods and References, please include data and publication citation for GSE141295. Please check our Author Guidelines for more information: <https://www.embopress.org/page/journal/17574684/authorguide#referencesformat>

- Please provide Reagents and Tools Table and uploaded it as a separate file. Structured Methods section includes Reagents and Tools Table followed by a Methods and Protocols section. More information on how to adhere to this format as well as downloadable templates (.docx) for the Reagents and Tools Table can be found in our author guidelines:

<https://www.embopress.org/page/journal/17574684/authorguide#structuredmethods>

An example of a paper with Structured Methods can be found here:

<https://www.embopress.org/doi/full/10.1038/s44320-024-00037-6#sec-4>

- Rename "Conflict of Interest" to "Disclosure Statement & Competing Interests" and place it after the "Acknowledgements". We updated our journal's competing interests policy in January 2022 and request authors to consider both actual and perceived competing interests. Please review the policy <https://www.embopress.org/competing-interests> and update your competing interests if necessary.

- Please rename "Availability of data and material" to "Data availability" and replace current sentence with "This study includes no data deposited in external repositories."

- Please correct the reference citation in the text and reference list. In the text a reference should be cited by author and year of publication. Include a space between a word and the opening parenthesis of the reference that follows. In the reference list, citations should be listed in alphabetical order. Where there are more than 10 authors on a paper, 10 will be listed, followed by "et al.". Also, please remove DOIs. Please check "Author Guidelines" for more information.

<https://www.embopress.org/page/journal/17574684/authorguide#referencesformat>

5) Funding: Please move it with "Acknowledgements".

6) Supplementary material: Please move "Supplementary Methods" to the Methods section in the main manuscript.

7) The Paper Explained: Please provide "The Paper Explained" and add it to the main manuscript text. Please check "Author

Guidelines" for more information. <https://www.embopress.org/page/journal/17574684/authorguide#researcharticleguide>

8) Synopsis: Every published paper now includes a 'Synopsis' to further enhance discoverability. Synopses are displayed on the journal webpage and are freely accessible to all readers. They include separate synopsis image and synopsis text.

- Synopsis image: Please provide a visual abstract as a high-resolution jpeg file 550 px-wide x 300-600 pixels high to illustrate your article.

- Synopsis text: Please provide a short standfirst (maximum of 300 characters, including space) as well as 2-5 one sentence bullet points that summarise the paper as a .doc file. Please write the bullet points to summarise the key NEW findings. They should be designed to be complementary to the abstract - i.e. not repeat the same text. We encourage inclusion of key acronyms and quantitative information (maximum of 30 words / bullet point). Please use the passive voice.

9) As part of the EMBO Publications transparent editorial process initiative (see our Editorial at <http://embomolmed.embopress.org/content/2/9/329>), EMBO Molecular Medicine will publish online a Review Process File (RPF) to accompany accepted manuscripts. This file will be published in conjunction with your paper and will include the anonymous referee reports, your point-by-point response and all pertinent correspondence relating to the manuscript. Let us know whether you agree with the publication of the RPF and as here, if you want to remove or not any figures from it prior to publication. Please note that the Authors checklist will be published at the end of the RPF.

10) Please provide a point-by-point letter INCLUDING my comments as well as the reviewer's reports and your detailed responses (as Word file).

I look forward to reading a new revised version of your manuscript as soon as possible.

Yours sincerely,

Zeljko Durdevic

Zeljko Durdevic
Senior Editor
EMBO Molecular Medicine

*** Instructions to submit your revised manuscript ***

1) a .docx formatted version of the manuscript text (including Figure legends and tables)

2) Separate figure files*

3) supplemental information as Expanded View and/or Appendix. Please carefully check the authors guidelines for formatting Expanded view and Appendix figures and tables at <https://www.embopress.org/page/journal/17574684/authorguide#expandedview>

4) a letter INCLUDING the reviewer's reports and your detailed responses to their comments (as Word file).

5) The paper explained: EMBO Molecular Medicine articles are accompanied by a summary of the articles to emphasize the

major findings in the paper and their medical implications for the non-specialist reader. Please provide a draft summary of your article highlighting

6) Author contributions: the contribution of every author must be detailed in a separate section.

7) EMBO Molecular Medicine now requires a complete author checklist

(<https://www.embopress.org/page/journal/17574684/authorguide>) to be submitted with all revised manuscripts. Please use the checklist as guideline for the sort of information we need WITHIN the manuscript. The checklist should only be filled with page numbers where the information can be found. This is particularly important for animal reporting, antibody dilutions (missing) and exact values and n that should be indicated instead of a range.

8) Every published paper now includes a 'Synopsis' to further enhance discoverability. Synopses are displayed on the journal webpage and are freely accessible to all readers. They include a short stand first (maximum of 300 characters, including space) as well as 2-5 one sentence bullet points that summarise the paper. Please write the bullet points to summarise the key NEW findings. They should be designed to be complementary to the abstract - i.e. not repeat the same text. We encourage inclusion of key acronyms and quantitative information (maximum of 30 words / bullet point). Please use the passive voice. Please attach these in a separate file or send them by email, we will incorporate them accordingly.

You are also welcome to suggest a striking image or visual abstract to illustrate your article. If you do please provide a jpeg file 550 px-wide x 300-600px high.

9) A Conflict of Interest statement should be provided in the main text

10) Please note that we now mandate that all corresponding authors list an ORCID digital identifier. This takes <90 seconds to complete. We encourage all authors to supply an ORCID identifier, which will be linked to their name for unambiguous name identification.

Currently, our records indicate that the ORCID for your account is 0009-0001-3593-2428.

Link Not Available

11) Include a Reagents and Tools Table as part of the Methods section, which can be downloaded from our author guidelines (<https://www.embopress.org/page/journal/17574684/authorguide#structuredmethods>)

Photos 400-800 DPI

*Additional important information regarding figures and illustrations can be found at

<https://bit.ly/EMBOPressFigurePreparationGuideline>. See also figure legend preparation guidelines:

<https://www.embopress.org/page/journal/17574684/authorguide#figureformat>

***** Reviewer's comments *****

Referee #2 (Remarks for Author):

I would like to commend the authors for their comprehensive and thoughtful responses to the reviewers' comments. They have addressed the concerns raised regarding the involvement of GLDC in IgA nephropathy (IgAN) by providing detailed explanations and additional data, which enhance the clarity and rigor of their findings. The inclusion of new clinical samples and the revised statistical analyses significantly strengthen their argument for GLDC as a potential therapeutic target.

Furthermore, the authors have improved the manuscript by revising the abstract for clarity, compiling primer sequences into tables, and enhancing figure legends, thereby improving the overall readability and interpretability of the figures. The incorporation of quantitative analyses and the provision of raw data further bolster the credibility of their results.

The addition of a mechanistic diagram in the discussion section effectively summarizes their findings and elucidates the proposed pathways involved in GLDC's role in mesangial cell proliferation.

Overall, the authors have successfully addressed the reviewers' critiques, and the revisions made to the manuscript render it suitable for publication. I believe this study will make a valuable contribution to the understanding of IgAN pathology and potential therapeutic interventions.

The authors addressed the remaining editorial issues.

16th Sep 2025

Dear Dr. Zou,

We are pleased to inform you that your manuscript is accepted for publication and is now being sent to our publisher to be included in the next available issue of EMBO Molecular Medicine.

Zeljko Durdevic
Senior Editor
EMBO Molecular Medicine
